# Sustainable care: How CSR shapes wellbeing in healthcare organizations in Beijing, Shanghai, and Guangzhou

Qinghua Fu[1]*, Belal Mahmoud AlWadi[2], Matac Liviu Marian[3], Rui Dias[4,5]

**1** Department of Business Administration, Moutai Institute, Zunyi, China, **2** Department of Basic Sciences (Humanities and Scientific), Faculty of Arts, Al-Zaytoonah University of Jordan, Amman, Jordan, **3** Faculty of Accounting and Management Information Systems, Bucharest University of Economic Studies, Bucharest, Romania, **4** Instituto Politécnico de Setúbal, Escola Superior de Ciências Empresariais, Setúbal, Portugal, **5** ISG |Business & Economics School, Linbon, Portugal

☙ Equal contributors.
* fqh15203972053@163.com

## Abstract

This article investigates the link between corporate social responsibility (CSR) perceptions of employees and employee burnout, from a sustainable development point of view, in the healthcare system of China. It fills the void in conventional literature by analyzing the indirect effect of CSR on the health and well-being of health workers aligning with SDGs focused on health and wellbeing. To be able to address the crux of healthcare professionals' burnout which can have far-reaching negative consequences for individual welfare and healthcare delivery, this research explores the linkages between CSR perceptions, employee burnout, happiness, psychological safety, and altruistic behavior. Data were obtained from 392 health care workers in three Chinese cities through a thrice-administered questionnaire that measures CSR perceptions of employees, burnout levels, happiness, safety perceptions, and altruistic motivations. It is seen that results show a strong link between CSR perceptions of employees and burnout reduction. Furthermore, the happiness and psychological safety of workers were expressed as mediators, with altruism playing the role of moderator. The above points highlight the need to adopt CSR strategies to promote employee well-being and combat burnout in the healthcare sector, which plays a vital role in global initiatives to attain SDGs related to good health and well-being, and sustainable development. Additionally, this research increases the debate on employee burnout based on their organization's CSR perceptions and positive psychology theory as a lens, bringing up CSR as the key factor in the achievement of sustainable development and the improvement of well-being within healthcare settings.

## 1. Introduction

### 1.1. Background

The pivotal role of employee mental health and well-being in an organization's success should not be ignored, as it is the basis of the performance and sustainability of business practices [1]. The difference is more significant in the services sector, where employees engage directly

**Data availability statement:** All relevant data are within the manuscript and its Supporting Information files.

**Funding:** This paper was financed by Instituto Politécnico de Setúbal.

**Competing interests:** The authors have declared that no competing interests exist.

with the customers, and they are, in essence, the service itself, rather than in the manufacturing sector, where machines and production are the focus [2]. In this environment, employees will have to do more than just follow instructions and they will be involved in the heart of the service delivery and satisfaction of customers. They assume special significance, therefore, as the deciding factor for organizational performance. Under such circumstances, the problem of employee burnout intensifies, and this is reflected through the negative consequences, such as psychological and psychical health issues of employees (like increase risks of depression and anxiety as well as chronic fatigue), which consequently, lead to considerable organizational disruption [3]. The changes in productivity, high turnover rates, low quality of service, and customer dissatisfaction which not only the organization's reputation but also its financial health. The particular element to be underlined is the complicated interaction between staff wellbeing and business performance, especially in healthcare service-driven organizations, which should be taken as necessary because of the concept of burnout considered to be a strategic and operational success feature. On a wider scale in healthcare this problem will have a negative impact on the efforts to achieve the Sustainable Developmental Goals (SDGs) such as good health and well-being in line with Goal 3, decent work and economic growth (Goal 8) and diminished inequalities (Goal 10) among others.

Because of these activities, healthcare providers act as the vital force or the lifeblood of societies and economies which help people to have a better life free of sickness and discrimination without being over-burdened. In spite of this, the issue of burnout among healthcare workers is a major issue that will prevent us from attaining these targets (SDGs) [4]. The exhaustion among healthcare workers hinders their own well-being while also SDG3 is undermined. This affects health workforce availability and results in inadequate care provision, which leads to increased health inequalities (SDG 10) and inefficiency or unproductivity in healthcare services (SDG 8). Coherently, employee burnout extends beyond the organizational interests to preclude global efforts towards achieving sustainable development and ensuring well-being for all which justifies the need to address the issue holistically using systemic interventions and policies aiming at the mental and physical health of healthcare professionals as a base for the achievement of the SDGs.

Interventions undertaken within organizations and include managing workloads, arranging work schedules, employee support programs, and the creation of a positive organizational culture is recognized for the possibility of preventing employee burnout [5]. These interventions aim at developing a healthy work environment where people are able to cope with the stressors that lead to burnout. Along with these interventions the field of corporate social responsibility (CSR) attracts attention with its potential to notably change employee outcomes [6]. CSR is defined as the actions and policies that companies undertake in order to address the effects of business operations on society [7]. Such efforts may include ethical business, utilization of resources, and interaction with the people in the society [8]. Employee burnout, on the other hand, refers to workplace stress that has not been properly addressed or managed [9]. This is evident in physical and emotional fatigue, a feeling of diminished productivity, and feelings of alienation or indifference to one's work. Understanding these key concepts is crucial for grasping the context of this study because this perspective may have a great potential to address the SDGs, including environmental sustainability (SDG 13), inequalities reduction (SDG 10) and economic growth promotion (SDG 8), and also help create a sense of purpose and belonging among employees. The fact that CSR is crucial from the SDGs standpoint is remarkable for it is a tool that integrates sustainable development in business operations and thereby contributes to the global goal of sustainable and inclusive growth.

Whereas a significant body of literature has already been accumulated on the positive impacts of employees' CSR perceptions on their engagement, satisfaction, and loyalty [10], the

potential of CSR in ameliorating negative outcomes at work, such as burnout, has not been explored significantly. Particularly in healthcare where the level of service, physicians provide, is directly influenced by their emotional health, the stress of working in this environment is especially high. Therefore, this study intends to fill this knowledge gap by furthering the debate on employee burnout in healthcare institutions, using the CSR framework, with a focus on SDGs. This study is aimed at showing how CSR perceptions at the level of employees can support healthcare workers in reducing burnout. It also intends to contribute to the broader research around sustainable development, employee well-being, and organizational success. This view not only answers a significant aspect of employee health and organizational research but also fits with the whole world's sustainable development targeting which emphasizes the interdependence of CSR, employee welfare, and the accomplishment of SDGs in the medical sector.

However, the influence of employees' CSR perceptions on employee outcomes is complex and multifaceted, contingent upon a mix of psychological and personal factors [7]. These include individual values alignment with organizational goals, perceived organizational support, and the psychological contract between employee-organization relationships among others. Echoing insights from Glavas [11], it becomes evident that without considering the mediating or moderating roles played by various psychological factors, CSR perceptions of employees cannot be comprehensively understood when it comes to influencing the attitudes or behaviors of employees. In this vein, we propose happiness & psychological safety as mediators for linking the CSR perceptions-employee burnout relationship. We choose to put emphasis on these mediators because we understand that although many studies have recognized the impact of employee happiness and psychological safety, on improving employee outcomes there is still a gap in our understanding of how these variables play a role in mediating the connection, between CSR perceptions and employee burnout. We choose to focus on these mediators because we know that although there is already evidence supporting that employee happiness and psychological safety lead to improved employee outcomes, we do not yet fully understand why and how CSR perceptions improve employee well-being through enhancing these aspects. Employee happiness gained from positive perception of CSR can be used to enhance engagement and satisfaction which could help to mitigate stressors that promote burnout. Moreover when ethical standards fostered by good CSR practices enable a psychologically safe environment then employees can voice their concerns about distress openly without fear of negative consequences hence decreasing chances for burnout. This proposal fills a significant gap in the literature on how perceived happiness and psychological safety as promoted by CSR serve as intervening variables in mitigating the adverse effects of burnout amongst healthcare workers. Besides, altruism is a major component of individual psychology, especially for careers with high service orientation that involve selflessness towards others' wellbeing [12]. In the health sector especially where the majority venture into this profession wanting patients to receive care [13], altruism becomes a central character trait in the healthcare profession. It is no wonder then that many individuals are attracted to healthcare careers by an intrinsic motivation to help others, which demonstrates a fundamental consonance with values promoted under CSR. The literature published in the last few years has identified a rising concern about the correlation between CSR perceptions and employee wellbeing especially in the high-pressure area such as the healthcare industry [14]. The healthcare industry is more amenable to CSR efforts because the jobs are stressful and sometimes emotionally demanding. In this regard, CSR is an effective way of combating burnout among healthcare personnel since it involves creating a supportive and ethical practice environment. The connection to the SDGs, especially SDG 3 and SDG 8 shows that CSR also has positive effects on society beyond just the organization [15]. Additionally, CSR is not only positively correlated with individual

outcomes but depends on several psychological and organizational factors [2,16,17]. This research will therefore seek to disentangle these interactions by examining how CSR contributes to the reduction of burnout through employee happiness and psychological safety. In this respect, the moderating influence of altruism will be discussed to establish its moderating capability in relation to CSR perceptions [18] and their outcomes on the health and wellbeing of employees. Thus in our present study, we propose altruism as a moderator acting within relationships hypothesized between CSR perceptions vis-à-vis employee happiness, psychological safety, and burnout.

## 1.2. Study segment

The study targets china's healthcare sector due to the rapidly changing economies, significant health reforms, and objectives of SDGs. China was chosen strategically as it is among the biggest in terms of size [12] in addition to facing immense challenges that give it an opportunity to offer more information on employee burnout compared to that obtained from most studies conducted in Global North countries. On the other hand, China's healthcare system is currently witnessing a complex path of rapid expansion, policy reform, and high patient load expectations. This is an opportunity different from what has been traditionally done in high-income countries like the USA or many European nations. Hence, this research tries to achieve several objectives by clarifying some theoretical gaps found within the existing literature.

## 1.3. Research gaps

The current study tends to bridge several research gaps in the existing literature. To begin with, whilst numerous studies have documented the positive effects of CSR on diverse aspects of employees' psychology, little empirical work has focused on how exactly CSR perceptions mitigate burnout amongst employees in the healthcare context especially in China. For that reason; this research will attempt to fill this gap by examining various complex channels through which CSR perceptions affect wellbeing leading to decreased levels of burnout among workers such as the mediating role played by psychological factors including happiness and psychological safety. Also, no empirical study (at least not known to us) demonstrates how SDG frameworks connect with CSR-related programs besides their impacts on staff performance. Particularly, this study acknowledges the healthcare industry's critical importance in the realization of SDG3 'good health & well-being,' and SDG 8 'decent work & economic growth' under the SDG framework; along with other related objectives, which provide a more comprehensive view on CSR beyond business outcomes [19]. Despite the extensive research on CSR perceptions and employee well-being, there remains a gap in understanding how these initiatives specifically impact burnout in the healthcare sector. Existing literature often focuses on general employee satisfaction and organizational commitment, leaving a critical gap in addressing the unique stressors faced by healthcare professionals. This study aims to bridge this gap by providing empirical evidence on the effectiveness of CSR perceptions in reducing burnout, thereby contributing to the broader discourse on sustainable development and organizational health.

Moreover, there is limited research work concerning the CSR perceptions-burnout correlation in the developing countries' healthcare sectors as the existing body of knowledge majorly focuses on the Global North thereby creating room for carrying this debate in a Global North's context. Thus, having chosen China's healthcare sector for this study, it broadens these dynamics so that CSR theories and practices may be generalized across different geographical and socio-economic locations. Finally, this study tends to examine how employee happiness,

psychological safety, and altruism can help to reduce burnout through employees' perceptions of CSR. Therefore, it contributes to the establishment of the theory regarding complex connections amid organizational policies, individual attributes, and employee outcomes. In terms of promoting employees' wellbeing as well as positioning CSR as a crucial determinant in achieving sustainable development in the healthcare sector. This research tends to fill critical theoretical gaps while giving practical recommendations for healthcare organizations willing to use CSR in improving the welfare state of their employees thus reaching SDGs.

The present research is a quantitative study and data for the analysis is gathered from healthcare workers in three large Chinese cities and data analyzed using structural equation modeling. The Positive Psychology Theory (PPT) acts as the theoretical lens through which the study examines how CSR perceptions can promote positive organizational behavior that boosts employees' happiness and psychological well-being. Through the combination of PPT in this study, it is possible to understand the processes by which CSR perceptions can help reduce burnout and support a healthy work climate.

The remainder of this paper is structured as follows: Section 2 presents the hypotheses formulated based on the theoretical framework and the literature review. Section 3 outlines the research method used in the study with regard to data collection and analysis. The analysis of the research study is discussed in Section 4 and Section 5 provides a discussion where the results are explained based on prior literature. Last of all, Section 6 presents the conclusion of the study.

## 2. Literature review

### 2.1. The underpinning theory

Our study finds Positive psychology theory (PPT) as the underpinning for the hypothesized relationships. Indeed the theory was developed by Seligman [20] and it aims at identifying positive aspects of human life as opposed to negative aspects of mental health. It also points out the essential components, which include positive emotions, engagement, relationships, meaning, and achievement (PERMA), in promoting human thriving [21]. Indeed, Seligman [20] approach was a radical change from traditional psychology where the major focus was on the abnormality of people. The PERMA model, which encompasses these five elements, offers a framework for understanding and measuring well-being, suggesting that these elements are essential for a fulfilling life.

Concerning the application of PPT in organizations, it assists in explaining how positive practices can enhance a work place environment [22]. Companies that adopt positive psychological factors may have the general goals of promoting positive organizational perspectives, personal strength, and effectiveness in workers. This is especially the case in sectors that are highly demanding such as the health sector due to the effects of burnout [23]. PPT can be implemented effectively with CSR strategies, as both initiatives promote the well-being of employees and a healthy work environment [24]. CSR initiatives such as ethical standards and community involvement are understood to relate to PPT as they provide meaningfulness for the employees. Organizational practices that are positive can help advance the levels of employee engagement, job satisfaction, and organizational commitment which helps to decrease burnout and increase the levels of psychological safety.

In this study, PPT is the theoretical foundation through which the effects of CSR on employee burnout, happiness, and psychological safety are examined [14]. In this respect, positive emotions, which PPT also emphasizes, are essential in countering stress and nurturing resilience to avoid burnout. Engagement, one of the PERMA factors, is supported by CSR activities that are relevant to employees' values hence improving employee satisfaction

at work. Also, ethical and inclusive CSR practices fosters good relationships and a psychological climate where employees feel protected and appreciated. It has also been established that there are various potential benefits that organizations may accrue when they align PPT with CSR programs; these include enhanced employee well-being [25], lower levels of turnover intentions [26], enhanced job performance, and greater innovation. With the promotion of a culture that enhances health and living, and embracing positive business practices, organizations are able to record high returns that go hand in hand with the overall improvement of the standard of living of the employees and the larger society. PPT provides insights into the ways by which CSR perceptions can prevent burnout and improve employees' quality of life and, therefore, is suitable for this research.

## 2.2. Hypotheses development

CSR has increasingly been linked to employee outcomes as a crucial element in fostering organizational commitment and satisfaction in the workplace [27]. These include programs incorporating ethical principles, ecological conservation, and improving social well-being among others [28]. This merging of personal and organizational values strengthens workers' ties to their jobs, resulting in more fulfilment, loyalty, and happiness. This accordance improves the working environment; hence it leads to better job performance and decreased turnover intentions and a number of consistent studies have shown a direct positive correlation between CSR activities and employee morale and productivity [29].

Regarding burnout at work among employees, their CSR perception is viewed as one approach that can prevent its occurrence. The concept of burnout is considered to be a serious issue within the healthcare system encompassing emotional exhaustion, depersonalization or cynicism and efficacy or low sense of accomplishment [30]. Therefore such CSR strategies as wellness programs, flexi-time systems, and employee assistance programs (EAPs) are real therapies for all roots of healthcare provider burnout. Occupational health schemes designed by firms that also offer flexible working hours for employees as well as consultancy services are highly effective with regard to managing job-related strains [31]. Additionally, by using CSR an atmosphere of respect for everyone is created where workers can raise their concerns without fear thus preventing burnout [32]. Other benefits that come along with this include safeguarding company image or developing staff pride thereby creating a mentally sound and motivated workforce.

Viewed through the lens of UN-SDGs, it implies that CSR transcends organizational boundaries to become part of a bigger global agenda. Alignment between SDGs and companies' corporate citizenship (CC) by focusing on health (SDG 3), economic growth (SDG 8), and inequality reduction (SDG10) demonstrates how CSR could be used as an instrument for addressing systemic inefficiencies in healthcare delivery system among others. Thus this shows the relation between organizational mechanisms, employee well-being/self-care behavior, and sustainable growth hence this emphasizes the strategic nature of CSR in the move towards a sustainable planet.

Our discussion revolves around positive psychology theory (PPT) which is based on fostering individuals' well-being along with their collective prosperity [20]. The CSR activities strengthen the work environment which sparks pleasurable experiences and personal development, among others. Such initiatives align with the PERMA model (Positive Emotions, Engagement, Relationships, Meaning, and Accomplishments), enhancing resilience and reducing burnout. More specifically CSR perceptions bring about employee satisfaction as they share common values and engage themselves in committed actions toward society through their roles at work this can evoke a positive affective experience at work leading to an

increased level of motivation [33]. Moreover, the approach addresses burnout stressors while promoting a workplace environment conducive to psychological well-being thereby enhancing enhanced job contentment and productivity.

**H1:** Employees' perceptions regarding the CSR engagement of their organization is negatively associated with employee burnout.

The level of employee's CSR engagement is directly associated with low levels of employee burnout.

The relationship between employee happiness and burnout is a hot topic in research. Numerous studies show that more happier workers are at work, and fewer likely they feel burnout [34]. An increase in job-related happiness may save an employee from stress-related burnout since all positive emotional energy produced by cheerful employees when stressed is available. Moreover, happiness is associated with resilience which helps employees to recover easily after a hardship occurs. This not only helps the individual but also enhances the general performance of the company [35]. Happiness is also a key mediator for CSR's impact on burnout. In case CSR aligns with the worker's values and has a strong influence within society, then employees will have a clearer purpose hence boosting pride at work as well. Happiness level among personnel raises their emotional wellbeing making them less vulnerable to burnout.

Previous research has established that CSR programs can effectively lower burnout as it promotes a healthy organizational culture that enhances employees' well-being [14]. Likewise, research findings show that CSR initiatives increase employee happiness because they help in achieving organizational and personal values of employees leading to feelings of purpose and gratification [8]. When viewed from the UN-SDGs perspective, the adoption of CSR strategies in which the happiness of the employees is paramount enables the organization to effectively be a part of global initiatives including the promotion of health among people through SDG 3 and decent job creation through SDG 8 [36]. Organizations can create a culture that values employees and thus promotes sustainable growth for each organization and the community as a whole. This presentation is based on PPT which gives the idea of the importance of happy, satisfied, and healthy organizations. This point of view helps us understand how resilience is increased by certain positive emotions such as happiness and decreases the chance of experiencing burnout. Hence, through CSR perceptions with PPT, organizations can achieve more employee happiness and suppress burnout.

**H2:** Employee happiness has a negative relationship with burnout.

**H3:** There exists a mediating mechanism of happiness in CSR-burnout relationship.

Psychological safety creates an environment where workers feel empowered to speak out, admit mistakes, and raise concerns without worrying about their self-image status, or even career [37]. Stress and anxiety are reduced when psychological safety is in abundance, singling out conditions that precipitate burnout [38]. In such circumstances, because stress as well as anxiety are reduced due to the presence of psychological safety that are known precursors of burnout, individuals can seek help if needed, take responsibility when appropriate, and have open conversations with colleagues thereby reducing levels of emotional exhaustion together with cynicism associated with burnouts. Accordingly, lack of psychological safety increases job pressures leading to more occurrences of burnout thereby underlining the necessity for nurturing psychological safety as an anti-burnout preventive measure [39]. We strongly advocate for psychological safety being designed as a preventative strategy against burnout.

Psychological safety perceptions from the employees can act as a mediator in the relationship between CSR perceptions and burnout. CSR initiatives generate a favorable and inclusive

work environment, encourage ethical conduct, and show respect for employees' well-being, which induce employees' psychological safety [40]. This emphasis on psychological safety as a way to lower burnout helps to create an avenue where workers can voice out their concerns, get help, and take part in meaningful social interactions without having fear of punitive measures [41]. The mediating role of psychological safety indicates how CSR initiatives contribute to a healthier work environment characterized by employee appreciation and protection, thus reducing the chances of experiencing burnout. Under UN-SDGs guidelines, building up a corporation's credibility in terms of fair labor practices is aligned with SDG 3 and SDG 8 [42]. Psychological safety is prioritized over other concerns since it takes into account not only individual wellness but also makes sure that there are sustainable resilient workplaces that support long-term organizational goals as well as general societal welfare. Hence, this perspective suggests the relationship between CSR performance at work; global sustainability goals; towards businesses creating conditions for psychological safety so as to prevent burnout. In this context, PPT applies to fostering positive workplace environments that promote psychological safety and well-being. Thus, PPT guides how the cultivation of situations where individuals feel psychologically safe reduces burnout rates. This implies that positive organization practices such as CSR might build resilience among employees against stressors leading to lessening of burnout.

**H4:** Employee perception of psychological safety negatively predicts burnout.

**H5:** There exists a mediating mechanism of psychological safety in the CSR perceptions-burnout relationship.

Altruism emerged as a significant moderator in the link between CSR perception and its impact on individuals [18]. For instance, those employees who are high on altruistic orientation are likely to gain enhanced levels of satisfaction and feel more psychologically secure when involved in CSR activities since such activities align with their self-regulated goal to help others [43]. Real-life examples show that organizations with solid CSR policies and standards have experienced improved positive results in altruistic employees. For example, Liu, Cherian [9] showed that CSR increased the level of both happiness and decreased burnout among employees in organizations that promoted altruism. When driven by good intentions for others' sake [44], altruism strengthens the association between employee happiness and CSR programs. This means that people who are highly altruistic at work will have increased job satisfaction and fulfillment when their organizations perform socially responsible acts because there is congruence between what they consider as important in life and the organization's policies [45]. This synergy can enhance the sense of happiness that arises from working in an enterprise with positive social impacts. Therefore, not only does altruism reinforce positively the CSR-employee happiness linkages but also weakens the likelihood of employee burnout [46] by promoting a sense of overall emotional well-being among other employees.

In the same way, altruism is a moderator that influences the relationship between CSR and psychological safety. Employees with altruism as a social tendency are more likely to perceive CSR initiatives as authentic steps towards social good, which in turn helps in building their sense of security and organization trust [47]. Consequently, this trust also contributes to creating a situation where employees with altruism feel safe enough to share ideas, voice out their concerns, and fully engage in company activities without any fear of being laughed or scorned at [48]. Thus, it can be concluded that CSR works optimally as long as its moderating effect genuinely promotes a psychologically safe workplace while at the same time protecting individuals from experiencing burnout due to an excessive desire to help others. The inclusion of altruism in the context of global sustainability goals focuses on how individual employee

action will affect the employee's outcomes concerning CSR. Moreover, increasing happiness among workers and ensuring their psychological well-being through proactive involvement in CSR functions is consistent with SDG 3 and SDG 8, indicating that wellness programs are essential for sustainable development. Based on PPT, this conversation shows how positive traits like altruism can have an exponential impact on the benefits of CSR initiatives. With this knowledge, we see how aligning personal values with the company's values can tip the scale in favor of optimal well-being and performance. When viewed through this lens, it is clear that altruism is not just a personal trait; it is also a tool that can be used to boost happiness, psychological safety, relieve burnout, and contribute to creating organizations that are truly sustainable for a longer period of time.

**H6:** Altruism moderates the relationship between CSR perceptions and employee happiness, such that the positive effect of CSR perceptions on employee happiness is stronger among employees with higher levels of altruism.

**H7:** Altruism moderates the relationship between CSR perceptions and psychological safety, such that the positive effect of CSR perceptions on psychological safety is stronger among employees with higher levels of altruism.

The Conceptual model of this study is shown in below Fig 1.

## 3. Methods

### 3.1. Study sector and context

Healthcare in China is recognized for its comprehensive and complex network that provides a service to the world's largest population. It has various health institutions, including hospitals which are public, and private, and common health facilities like community health centers and clinics. Healthcare is undergoing the most drastic transformation by which the government

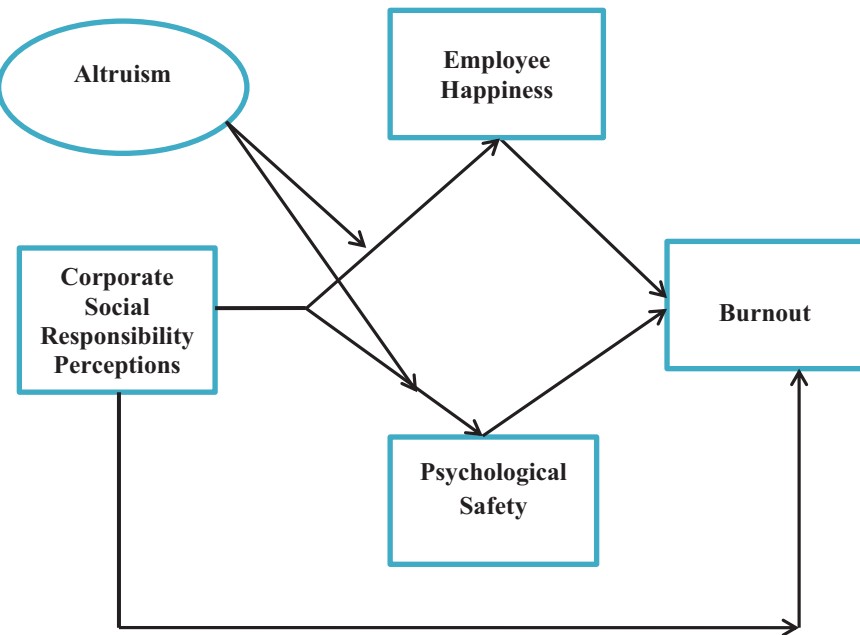

**Fig 1. The Conceptual Model.**

aims to make healthcare delivery improvements, availability, quality, and affordability. These reforms put the accent straight on CSR which specifically deals with the role of the healthcare system in helping people become healthy and in integrating environmental sustainability and ethical governance.

This study focused on three key cities such as Beijing, Shanghai, and Guangzhou whose selection was based on their strategic positioning in China's healthcare landscape as political economic, and health innovation hubs. These cities were chosen for their strategic positioning in China's healthcare landscape as political, economic, and health innovation hubs. In terms of medical policy formation and reform, Beijing is among the leading cities. Shanghai exhibits the most advanced medical technologies through its role as a financial center while at the same time experiencing significant growth in the private hospital industry. For a long time, Guangzhou has been famous for medical research and educational institutions promoting the development of medical standards and practices. This selection of cities offers a holistic picture of the CSR practices in various healthcare contexts in China, improving the study's validity and generalizability

. Since the aim was to have a comprehensive analysis of CSR, the participant organizations were selected based on the existence of organized, well-developed, advanced, and recognized CSR programs. However, we admit that this criterion might limit the generalization of the findings to the whole healthcare industry. In light of this, it is recommended that future studies incorporate organizations that have different degrees of CSR involvement to get improved insights. The administrations of these hospitals were requested to take part in this data collection activity, pointing to the mutual benefits for both academia and the healthcare sector. Those hospitals that responded positively were approached to begin the data collection exercise.

### 3.2. Ethics statement

The Helsinki Declaration's guiding principles were followed in this study to ensure ethical standards were upheld during research. It necessitated: (i) obtaining written informed consent from all respondents; (ii) ensuring that the identities of respondents remain anonymous; and (iii) allowing participants to withdraw at any time without any consequences. Moreover, the Ethics Committee of Maoutai Institute approved the data collection protocols of this study (DB85P4S24R).

### 3.3. Participants and data collection

Data from hospital staff were collected in three waves, starting from June 5, 2023, and concluding on July 20, 2023. The study used a prospective design with data collected in three waves at two-week intervals. The use of two weeks break in administering the survey waves aimed at reducing recall bias and capturing temporal changes in attitudes and perceptions in a relatively short period thus increasing the validity of the findings [49,50]. This design enabled us to capture active involvement with CSR initiatives and the consequent outcomes for employees. Moreover, we have also provided a detailed section of the discussion on possible sources of non-response bias. To reduce the effect of non-response bias, measures like follow-up reminders and maintaining the anonymity of the respondents were used. However, it is recognized that non-response bias may still be a problem, and it is identified as one of the limitations of the current study. The first wave began on June 5, followed by the second wave on June 19, and the third wave started on July 3. In the first wave, information was captured on CSR perceptions and employee altruism (AL) to establish initial links between organizational practices and individual characteristics. The second wave aimed at assessing

employee happiness (HP) and perceptions of psychological safety (PS), capturing the mediating variables under consideration. Accordingly, the last wave entailed the measurement of the employee's burnout (EB) so that the relationships between CSR perceptions, altruism, happiness, psychological safety, and burnout could be examined over time. To enhance the analysis of the data, structural equation modeling (SEM) with a high level of fit was employed to enable the determination of the structural interconnections among the variables. To confirm the constructs and the hypothesized relationships, we used a strong SEM measurement model and a structural model.

### 3.4. Measures

The five-point Likert scale was composed of one to five where disagree to agree was represented through the adaptation of well-acknowledged sources that gathered data from healthcare organization workers. The survey contained CSR perceptions, EB, HP, PS, and AL as the variables being measured. For instance, six items were used to measure employee perceptions regarding the general CSR engagement of their organization, with items such as "my hospital aims at protecting and enhancing environment". These items were extracted from the seminal work by Turker [51]. Indeed, the scale developed by Turker [51] consists of four dimensions which are assessed by 18 questions and it targets CSR to social and non-social stakeholders, employees, customers, and government. For the purpose of this study, a shortened version with six items focusing on the general CSR involvement of an organization was used. These items include: my hospital participates in activities aimed at protecting and improving the quality of the natural environment (1), makes investments to create a better life for future generations (2), implements special programs to minimize its negative impact on the natural environment (3), targets sustainable growth which considers future generations (4), supports non-governmental organizations working in problematic areas (5), and contributes to campaigns and projects that promote the well-being of society (6). This was done deliberately to reduce the time taken to complete the survey and to concentrate on the perception that the respondents had of CSR in general and not on the dimensions of CSR. Other prior works like have also employed a similar approach successfully to measure the general CSR engagement of an organization using a lesser number of items from this scale [2,6,7]. This method has been proved to be efficient for capturing CSR constructs while keeping the survey short and engaging for the respondents. Consequently, by taking a broad approach by examining the general level of CSR engagement, we were able to capture the overall effect of CSR on employee outcomes which is in line with our study goals. On the EB construct, there were seven items adapted from Kristensen, Borritz [52] which include statements like "I feel worn out after work". Similarly, García del Junco, Espasandin Bustelo [53] scale had five questions for measuring HP; for example, 'the hospital's climate is good'. Furthermore, PS was evaluated using seven items provided by Edmondson [54] like "no one would undermine my efforts here". Lastly, AL comprised three items according to Dotson, Dave [55]: "I joined healthcare to help others." Table 1 includes the variables and their operationalization

### 3.5. Method bias

Taking theoretical and empirical steps helped us avoid biases such as social desirability bias (SDB) and common method variance (CMV). Conceptually, all respondents were assured of their anonymity as well as confidentiality so that they could be free and give unbiased responses without any pressure from others [57,58]. Additionally, the questions for different constructs were varied in scale and randomly placed across the questionnaire to avoid predictability and peripheral cues that can lead to CMB. Moreover, a Harman single-factor

test was conducted as an empirical check for CMB [59,60]. The results of this analysis showed no salient single factor, suggesting that there was no significant problem with CMB in our dataset.

We utilized a-priori sample size calculator specifically designed for structural equation modeling (SEM) covering partial least squares (PLS-SEM) analysis in order to determine the right sample size for our study. PLS-SEM has been found particularly useful in exploratory studies aimed at theory building and when dealing with complicated models, as well as non-normal data, therefore, it is important to use this calculator that gives the most accurate estimates of the possible smallest number of samples to achieve a great statistical power for structure analysis. The recommended total sample size by this calculator was 377 taking into account such factors as indicators, latent constructs as well as expected effect sizes which were required to perform a robust structural analysis. Since high response rates are difficult to attain in survey research [61,62], we submitted 550 questionnaires among hospital personnel. Three waves of data collection yielded a total of 392 valid responses after deleting incomplete answers thus giving us about seventy-one percent of the actual response rate which is within the acceptable range whereas Table 2 below gives an overview of our sample's demographics.

## 4. Results

In Table 3, CFA results are discussed focusing on different variables under scrutiny. Factor loadings measure how strongly items relate to their underlying factors/constructs. When there is high factor loading coupled with t-statistics greater than ±1.96 (significant at 5%), it demonstrates a substantial relationship. For instance, items, attached to AL like, AL1 (0.873), AL2 (0.841), and AL3 (0.88) all have a high factor loading, which shows that they have a strong relationship with AL. However, there is a small issue with one item of EB as EB4 had a weak factor loading and hence excluded from the further analysis. Fig 2 represents the measurement model of our study

Looking at Cronbach's alpha and rho_A values, one can conclude there exists a high internal consistency in each of our variables because all their values are above 0.70. The composite reliability (CR) values exceed 0.70 thus showing acceptable measures [63,64]. AVEs also for each construct range from 0.50 upward which is more than enough to confirm their validity. Given that all the AVEs surpass 0.50, we can say that every latent variable sufficiently meets reliability and validity requirements [65,66]. Additionally, we assessed the possibility of multicollinearity among latent variables by calculating the Variance Inflation Factor (VIF) for each construct. All VIF values are below the threshold of 5, indicating that multicollinearity is not a concern in our model [67]. For more detail, we refer the readers to Table 4.

**Table 1. Operationalization of the variables.**

| Variable | Operational Definition | Source |
|---|---|---|
| CSR perceptions | General engagement of the organization's CSR initiatives | Turker [51] |
| Employee Burnout | Levels of emotional exhaustion and depersonalization | Maslach and Leiter [56] |
| Employee Happiness | Overall job satisfaction and sense of well-being | García del Junco, Espasandin Bustelo [53] |
| Psychological Safety | Perception of a safe environment to take risks and voice concerns | Edmondson [54] |
| Altruism | Tendency to help others and derive satisfaction from such actions | Dotson, Dave [55] |

**Table 2. Demographic information.**

| Demographic Variable | Frequency | Percentage (%) |
|---|---|---|
| Gender | | |
| Male | 152 | 38.8 |
| Female | 240 | 61.2 |
| Age | | |
| 20–30 | 112 | 28.6 |
| 31–40 | 176 | 44.9 |
| 41–50 | 72 | 18.4 |
| 51+ | 32 | 8.1 |
| Department | | |
| Nursing | 152 | 38.8 |
| Emergency Services | 88 | 22.4 |
| General Medicine | 64 | 16.3 |
| Psychiatry | 56 | 14.3 |
| Support Services | 32 | 8.2 |
| Years of Experience | | |
| <5 years | 120 | 30.6 |
| 5–10 years | 152 | 38.8 |
| >10 years | 120 | 30.6 |
| Job Role | | |
| Clinical Staff | 280 | 71.4 |
| Administrative Staff | 112 | 28.6 |

As can be seen in Table 5, the square root of AVE and inter-correlations for each construct are presented which indicates that there is high discriminant validity. The square root of AVE for each construct on the diagonal should be higher than off-diagonal values to show proper discriminant validity [68]. For example, AL has a value of 0.865, which surpasses its correlations with CSR perceptions, EB, HP, and PS. Another instance is CSR's value of 0.778 which is higher than other constructs it correlates with thus confirming that the measures are distinct from one another.

We used partial least squares structural equation modeling (PLS-SEM) using SMART-PLS for data analysis which is one of the preferred choice among a range of behavioral scientists for data analysis [69–71].

SMART-PLS is also effective in handling high multicollinearity between the predictor constructs and can accommodate them within the model. This capability is crucial for our research because happiness and psychological safety are two different concepts that are different, yet closely connected (it is also evident from the value of correlations between HP and PS which is 0.527). The previous researchers [72,73] have suggested employing sequential mediation as a preferred choice in such situations, our choice to maintain these constructs as independent mediators is deliberate. The correlation of 0.527 between HP and PS in this study indicates a moderate relationship, but this does not necessarily imply causality or the need for a sequential mediation model. By treating these variables as separate constructs, we can better isolate their unique contributions and predictive power regarding burnout, preserving the theoretical integrity and clarity of our model. A plethora of previous researchers have done it in recent studies [49,74]. Our choice of SMART-PLS for data analysis ensures that our model is optimized for predictive accuracy and explanatory power [75], aligning with the primary objectives of our research.

**Table 3. The CFA results.**

|  | Items | Standard Deviation | T Statistics |
|---|---|---|---|
| AL1 ← AL | 0.873 | 0.015 | 57.018 |
| AL2 ←AL | 0.841 | 0.019 | 43.946 |
| AL3 ← AL | 0.88 | 0.013 | 67.36 |
| CSR1 ←CSR Perceptions | 0.822 | 0.028 | 29.59 |
| CSR2 ← CSR Perceptions | 0.827 | 0.027 | 30.10 |
| CSR3 ← CSR Perceptions | 0.692 | 0.042 | 16.344 |
| CSR4 ← CSR Perceptions | 0.849 | 0.02 | 42.238 |
| CSR5 ←CSR Perceptions | 0.774 | 0.031 | 25.114 |
| CSR6 ← CSR Perceptions | 0.69 | 0.043 | 16.121 |
| EB1 ← EB | 0.698 | 0.149 | 4.679 |
| EB2 ← EB | 0.836 | 0.161 | 5.182 |
| EB3 ← EB | 0.740 | 0.173 | 4.263 |
| EB5 ← EB | 0.700 | 0.114 | 6.145 |
| EB6 ← EB | 0.806 | 0.117 | 6.887 |
| EB7 ← EB | 0.732 | 0.128 | 5.731 |
| HP1 ← HP | 0.887 | 0.013 | 66.075 |
| HP2 ← HP | 0.815 | 0.025 | 32.709 |
| HP3 ← HP | 0.725 | 0.030 | 23.951 |
| HP4 ← HP | 0.783 | 0.023 | 33.385 |
| HP5 ← HP | 0.72 | 0.03 | 24.108 |
| PS1 ← PS | 0.711 | 0.035 | 20.39 |
| PS2 ← PS | 0.864 | 0.017 | 49.396 |
| PS3 ← PS | 0.851 | 0.018 | 46.445 |
| PS4 ← PS | 0.824 | 0.018 | 45.61 |
| PS5 ← PS | 0.785 | 0.031 | 25.463 |
| PS6 ← PS | 0.771 | 0.031 | 25.076 |
| PS7 ← PS | 0.722 | 0.029 | 24.647 |

Table 6 presents a structural analysis of our proposed relationships as hypothesized in the conceptual model of our research. CSR perceptions affect EB directly (−0.098) at p = 0.008 level; CSR perceptions also influence HP (0.158) and PS (0.232) significantly both having a p-value = 0.000, indicating its positive effect on these variables. Similarly, HP and PS also significantly predicted EB (−0.163 and −0.178 with p values < 0.05). The CSR perceptions -> HP -> EB pathway reveals a mediation effect (−0.026) by positioning HP as a mediator. PS's relationship with EB is also significant (−0.178, p-value: 0.022), and the path CSR perceptions -> PS -> EB confirms PS as another mediator with a coefficient of 0.041. The path CSR*AL1 -> HP -> EB is also significant (0.037, p-value: 0.006), showing that AL alters the connection linking CSR perceptions and EB via HP. As well as, the relationship between CSR perceptions*AL2 and PS is also significant (0.055, p-value: 0.002); it means that AL moderates through PS. Fig 3 below represents our structural model showing various relationships.

## 5. Discussion

In our present study, we provide empirical evidence about CSR perceptions, employee burnout, happiness, psychological safety, and altruism dynamics within the healthcare sector in China. The first hypothesis posited that "CSR perceptions will have a negative direct relationship with employee burnout." This finding supports prior research that suggests that CSR perceptions

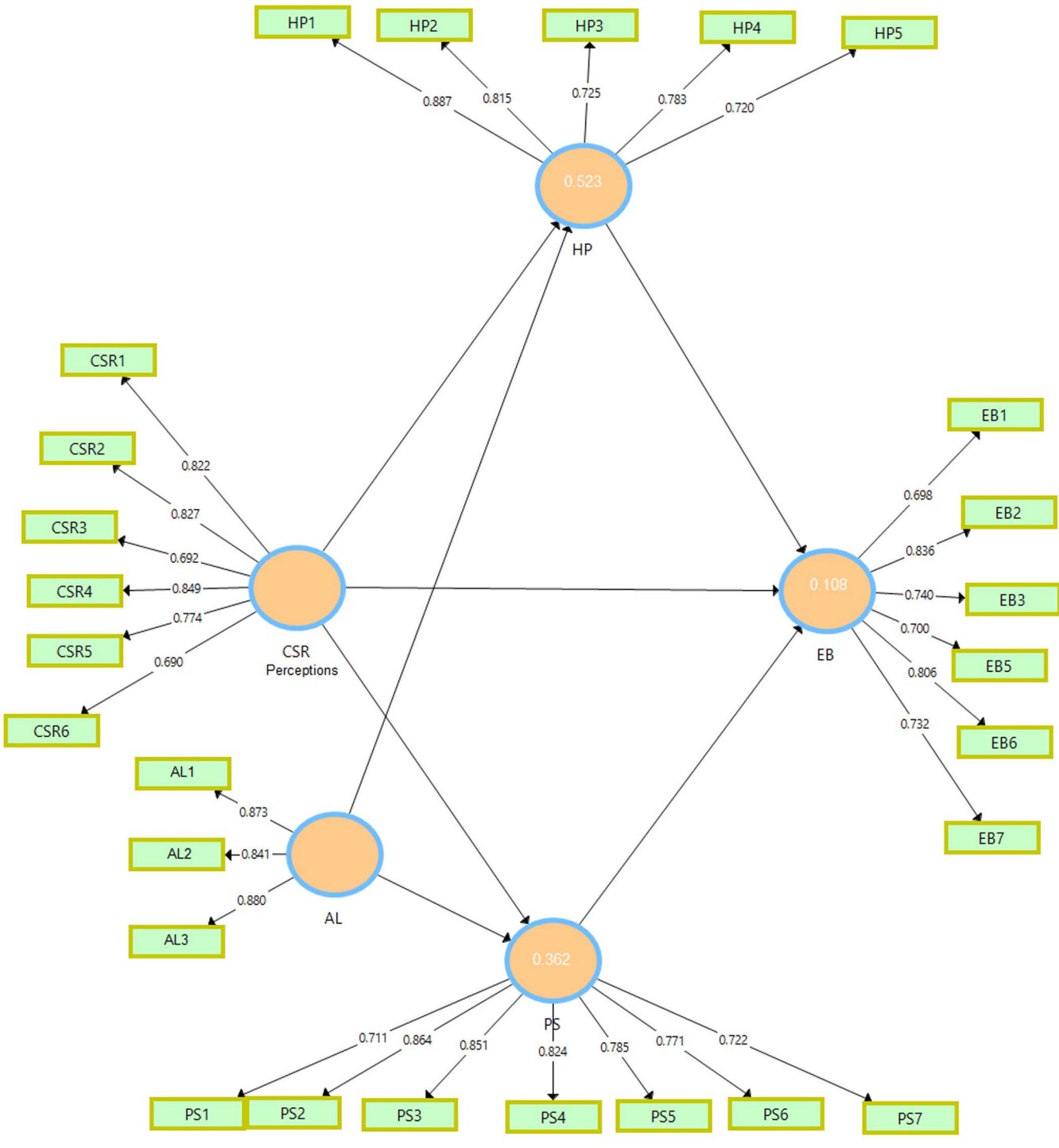

**Fig 2. The measurement model.**

decrease the effects resulting from burnout by promoting supportive organizational climates that prioritize employee well-being [32,76]. Consequently, the positive effects of CSR perceptions on employee happiness and psychological safety highlight its pivotal role in creating employees who value their respective organization's values. These findings support positive psychology theory

Table 4. Reliability and validity.

| | Cronbach's Alpha | rho_A | Composite Reliability | Average Variance Extracted |
|---|---|---|---|---|
| AL | 0.831 | 0.833 | 0.899 | 0.748 |
| CSR Perceptions | 0.873 | 0.894 | 0.902 | 0.606 |
| EB | 0.855 | 0.932 | 0.887 | 0.568 |
| HP | 0.846 | 0.854 | 0.891 | 0.622 |
| PS | 0.900 | 0.909 | 0.921 | 0.627 |

Table 5. Discriminant validity.

| | AL | CSR | EB | HP | PS |
|---|---|---|---|---|---|
| AL | 0.865 | | | | |
| CSR Perceptions | 0.163 | 0.778 | | | |
| EB | 0.153 | −0.108 | 0.754 | | |
| HP | 0.686 | 0.245 | −0.118 | 0.788 | |
| PS | 0.510 | 0.282 | −0.111 | 0.527 | 0.792 |

Table 6. Hypothesis results.

| | Original Sample | Standard Deviation | T Statistics | P Values | 2.50% | 97.50% |
|---|---|---|---|---|---|---|
| CSR Perceptions → EB | −0.098 | 0.044 | 2.227 | 0.008 | −0.202 | −0.066 |
| CSR Perceptions → HP | 0.158 | 0.039 | 4.055 | 0.000 | 0.097 | 0.246 |
| CSR Perceptions → PS | 0.232 | 0.045 | 5.155 | 0.000 | 0.131 | 0.305 |
| HP → EB | −0.163 | 0.072 | 2.264 | 0.000 | −0.232 | −0.111 |
| PS → EB | −0.178 | 0.085 | 2.094 | 0.022 | −0.211 | −0.102 |
| CSR Perceptions → HP → EB | −0.026 | 0.012 | 2.167 | 0.000 | −0.038 | −0.012 |
| CSR Perceptions * AL1 → HP → EB | 0.037 | 0.014 | 2.643 | 0.006 | 0.011 | 0.046 |
| CSR Perceptions → PS → EB | −0.041 | 0.017 | 2.412 | 0.000 | −0.059 | −0.022 |
| CSR Perceptions * AL2 → PS → EB | 0.055 | 0.021 | 2.619 | 0.002 | 0.019 | 0.066 |

which asserts that positive organizational practices lead to flourishing or optimal functioning. Moreover, our results suggest that employee happiness and psychological safety (H2 and H4) were negatives for predicting employee burnout. These results concur with previous studies showing both happiness and psychological safety as essential to mitigating the negative impact of stressors within an organization leading to burnout [34,38]. This idea corresponds to Fisher [77] belief who stated that reduced burnout rates are naturally associated with greater workplace wellbeing due to favorable emotional experiences being protection mechanisms against work-related stressors. Moreover, on psychological safety, Edmondson [54] argues such environments enable members to take risks and they should be safe enough for themselves not to feel threatened because eventually there might be no fulfillment thus leading them down a path toward breakdown especially when it is characterized by exhaustion. In addition to supporting the empirical evidence on the critical roles of happiness and psychological safety in enhancing workplace well-being, these findings suggest that organizations should strive to create environments that foster these positive psychological states to combat burnout effectively. Building on and going beyond the insights of these past studies, our work sheds light on how promoting employees' happiness and creating psychological safety can be critical organizational imperatives in preventing burnout and improving the overall employee wellbeing.

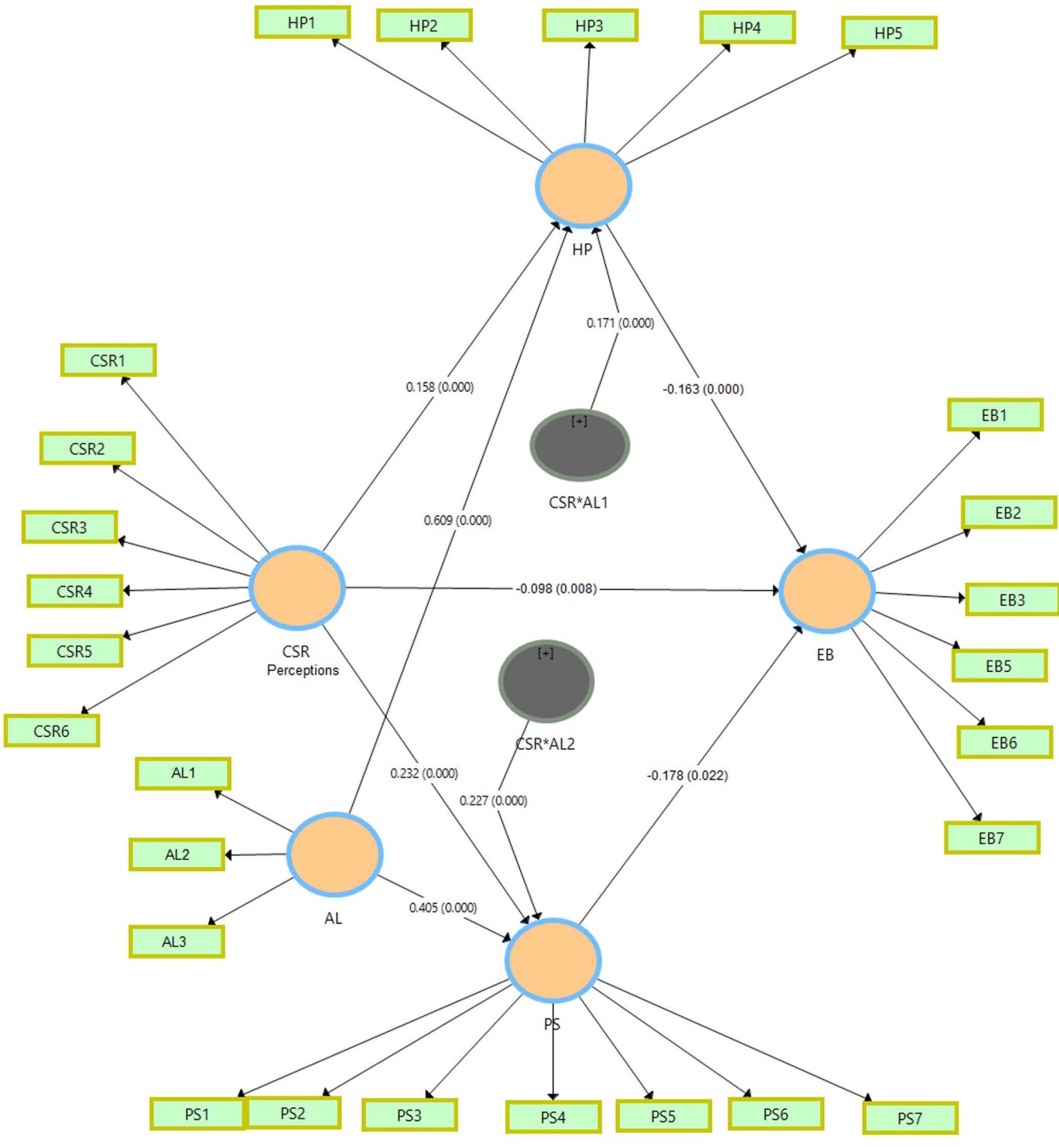

**Fig 3. The structural model.**

Another finding indicated in H3 "There will be a negative effect of CSR on employee burnout mediated by happiness." This finding also agrees with previous work as it suggests that organizations can mitigate elements leading to burnout through efforts such as promoting a positive organizational climate and commitment to worker's welfare [78]. Other studies

have confirmed the mediating effect of employee happiness on reducing burnout [79]. Our study also confirmed the mediating mechanism of psychological safety in the CSR perceptions-burnout relationship. This finding supports the notion that CSR perceptions contribute to employee well-being not only directly but also by enhancing factors known to buffer against burnout. The moderating role of altruism in the relations between CSR perceptions and both happiness and psychological safety, however, points to variance among individuals regarding their propensity for CSR activities. Thus, personnel who are truly grateful for helping others may be more gratified and feel safer resulting from the company's projects related to these employees' basic needs that eventually cut job burnout. In this respect, individual differences play a vital role in understanding how CSR perceptions affect employee well-being; therefore implying a more sophisticated approach to understanding how and why such practices impact personnel outcomes.

Also, the moderating effects of altruism (H6 and H7) on the relationships of CSR perceptions with happiness and psychological safety imply that personal-level altruistic tendencies can affect the success of CSR activities. This implies that employees innately committed to helping others would derive more satisfaction and security from CSR activities, therefore, experiencing less burnout. Such results stress the need to take into account personal characteristics when analyzing the relationship between CSR perceptions and employees' wellbeing, implying a more complex view of the dynamics, which lie behind the connection between CSR perceptions and employee outcome.

Finally, results from this study are consistent with positive psychology theory's contentions that the use of positive organizational practices such as CSR can lead to increased employee happiness, wellbeing, and psychological safety while reducing burnout levels. The happiness-mediation and psychological-safety mediating roles in addition to altruism-moderation explain some non-trivial aspects of the complex reciprocity relationship between organizational initiatives and personal employee outcomes. Our research identifies how important it is for companies to create a healthy work place as well as other features needed for good organizational culture; hence it emphasizes the significance of positive psychology theory as a framework upon which policies regarding staff well-being should be based in order to enhance sustainable development within the organization. Moreover, this perspective helps people achieve greater quality-of-life satisfaction at work while also addressing principal objectives like sustainability and business success. The results of our study are evidence of the effectiveness of CSR perceptions in mitigating burnout and promoting well-being, happiness, and psychological safety among employees. These findings are similar to previous studies that have highlighted a positive relationship between CSR and different employee outcomes [27]. For instance, earlier research has shown that CSR perceptions drive the creation of a positive organizational environment that translates to employee commitment and satisfaction [29]. When organizations adopt an ethical climate and social responsibility, they foster good psychological states among the employees [31]. Therefore, our study enhances this line of literature by extending the understanding of how and why CSR perceptions reduce employee burnout through the mediating factors of happiness and psychological safety. The positive psychology theory affirms our results in the sense that positive work characteristics, including CSR perceptions, promote positive psychological experiences that can counterbalance the negative impacts of workplace stressors. A moderate positive correlation was found between the levels of happiness and psychological safety in our study (r = 0. 527) which indicates that although these factors are related, they both have significant positive effects on decreasing burnout. This differentiation is necessary to find the best strategies that will focus on particular elements of workers' health.

Furthermore, the extension of the previous research by integrating the moderator of altruism [80,81] helps to deepen the knowledge of the role of individual characteristics in the

effectiveness of CSR activities. Organizations that have high levels of CSR activities are likely to be more appealing to employees with high altruistic tendencies because such activities are actually in line with the employees' self-interest in helping others [45]. This implies that organizations need to complement the execution of CSR programs with the promotion of altruistically appropriate norms.

Based on our research, it is possible to conclude that CSR perceptions are positively associated with a decrease in the burnout level of employees and an increase in their happiness and psychological safety at work in the context of the Chinese healthcare industry. This can be explained by the Positive Psychology Theory which underlines the role of positive organizational practices in increasing the level of employees' well-being. The two mediators, namely; happiness and psychological safety, and the moderator; altruism give a clear picture of how CSR perceptions impact burnout. Such findings are useful for enhancing the interpersonal and inter-organizational processes that impact employees' quality of life.

## 5.1. Theoretical implications

Our research makes a substantial contribution to existing literature on CSR and its effects on employee outcomes, more so in relation to the Chinese healthcare sector. We examine the direct association of CSR perceptions with employee burnout, and how happiness and psychological safety mediate this relationship, as well as altruism moderating it. This way we bridge some gaps in the existing literature. To start with, our findings point out that there is a direct link between CSR perceptions and employee burnout other than just job satisfaction or organizational commitment which are common outcomes of CSR perceptions. The implication here is that burnout focused environment can help to guard against chronic stress at work. In the healthcare sector particularly where levels of stress are high. Secondly, we establish that happiness and psychological safety could serve as mediators between CSR perceptions and burnout. This reveals how positive mental states can be employed to reduce burnout incidences. We conclude that taking action to improve these two aspects would result in healthier employees overall. Thirdly, it is crucial to recognize individual differences when measuring outcomes of CSR initiatives because they shape those outcomes quite a bit. Our study implies that if employees have high levels of altruism then CSR efforts could be more effective for them personally. By personalizing approaches towards help instead of using broad strokes we can value everyone individually. Lastly, our research addresses an existing gap between the perspective of SDGs and CSR initiatives within organizations, and the literature on both topics together. By demonstrating how CSR initiatives can help with employee wellbeing, reduce burnout, and enhance happiness and psychological safety we provide empirical evidence about SDG alignment with several business practices like SDG 3 (good health and wellbeing) and SDG 8 (decent work and economic growth). Our contribution is vital in furthering conversations about global sustainability objectives through responsible business practices, making them crucial for scholars & practitioners alike who dabble in CSR.

## 5.2. Practical implications

The results of this study are extremely important for China's healthcare industry. Dealing with employee burnout is a health sector concern, impacting not only the welfare of its workforce but also on level of patient care. Our research indicates how CSR helps to alleviate symptoms related to burnout. It highlights how such initiatives can create a more supportive and engaging work environment for employees. This becomes extremely necessary when the demand and stress levels among workers are at their peak. Application of the CSR practices may assist in building happiness and psychological safety at work places which are crucial in

overcoming burnout. Furthermore, our findings show that hospitals should primarily adopt CSR as their organizational culture and operations model rather than any other practice within them including making policies and practices that put employee wellbeing first; having support systems in place to keep work-life balance possible; providing opportunities for advancement, etc. The company management should understand that they have not been acting right towards their workers by taking this direction since it shows that they value those whose health will have a direct impact on business status or profits. Instead, firms need to always indicate through their CSR strategy how they provide a healthy working environment all time long that is flexible and supportive as we have already mentioned above. Besides, our study underlines the critical significance of individual differences like altruism in determining CSR-employee outcomes relationships. This implies that different values and motivations should be acknowledged so as to exploit these benefits fully in relation to various individuals under employment who form part of the workforce while appreciating different values within them. Personalizing their efforts towards attaining employees' personal values systems plus career aspirations will surely result in remarkable improvements regarding upward spirals about impacts from CSR as far as workforce wellness is concerned.

In broader terms, this study is relevant to the discussion of how CSR activities in the healthcare industry support SDGs within the larger SDG framework. Chinese healthcare companies can achieve SDG 3 through initiatives that address employee burnout. Also, these organizations can promote business-oriented targets such as SDG 8, by making their work-force operate in conditions where better employment facilities are available with improved economies too. Therefore, our research not only brings out the practical implications of CSR in the healthcare sector but becomes significant with regard to global sustainability efforts, thus emphasizing the crucial role of healthcare enterprises in creating a better, more sustainable world. The broader implications of our study are important at the level of organizational practice and for policy making. Leveraging our results, organizations should consider embracing CSR as a strategic component aimed at improving employees' well-being and productivity. The findings of this study suggest that by investing in CSR activities, organizational outcomes are enhanced because of the improved work environment. Policy makers can also use these insights to champion CSR as part of the corporate sustainable development goal and practices to ensure that the companies adopt appropriate behavior that will be in the best interest of the employees as well as the larger society.

## 5.3. Limitations and future research directions

Although our research contributes greatly to the understanding of the CSR perceptions -employee burnout relationship in the context of the healthcare industry in China; however, it is not without limitations for future studies. To illustrate, this study was only conducted in major cities where China's healthcare is concentrated which does not represent the experiences of healthcare workers in rural and less developed areas with different implications of CSR perceptions on burnout. Additionally, the use of cross-sectional data limits our ability to make causal inferences hence we need longitudinal studies that cover temporal dynamics between CSR and employee outcomes. Cultural factors could act as mediators between CSR and employee wellbeing but other countries can also be studied. Qualitative methods would also provide deeper insights into employees' subjective experiences with regard to these initiatives thereby leading to a better understanding of their influence on burnout. For the suitability in small sample sizes and exploratory research, SEM was conducted in this study using SMART-PLS. Unfortunately, SMART-PLS does not offer model fit indices (e.g., CFI, TLI, GFI, RMSEA), thus preventing us from conducting a complete model equivalence evaluation. A recognized limitation is that model fit testing would add rigor to evaluating collective

relationships among variables. Although SMART-PLS enabled iterative model refinement and investigation of mediation and moderation effects via bootstrapping, it has relative lower statistical power than the covariance-based SEM approaches like AMOS. This limitation should be addressed in future studies through utilization of larger datasets and advanced SEM techniques including model fit indices as well as sequential mediation frameworks to more robustly evaluate interconnections between variables. Results from this exploratory study contribute to understanding how CSR perceptions relate to burnout via mediators. To increase the validity and generalizability of the findings, however, there may be a need to transition to confirmatory analysis in future research.

Further, it is crucial to note that there are certain issues that were beyond the scope of our research and need to be addressed concerning CSR implementation. CSR best practices must be accompanied by significant organizational commitment and resources for it. Some of the difficulties that firms face when implementing CSR strategies include; a lack of integration of CSR with the company's objectives, employee involvement, and achieving sustainability of CSR. In addition, the study is mainly based on urban hospitals, which can partially reflect the problem faced by health care organizations in the rural area. Further study should be conducted to prove the different impacts of CSR in rural areas to have broader research on the influence of CSR. On a final note, although happiness and psychological safety, are conceptually distinct, are moderately correlated. These factors were modeled as independent mediators in this study, allowing the unique contributions of each of these factors on the CSR perceptions-burnout relationship to be examined. On the one hand, happiness represents emotional wellbeing arising from practices of positive organizations; on the other hand, psychological safety represents an interpersonal climate that promotes openness and mitigates anxiety, both of which are necessary to understand multidimensional effects of CSR. This perspective assesses their individual impacts, but does not assess sequential mediation or interdependencies. Simplicity was retained in the model due to limitations in sample size and analytical capabilities of the software used in this study. Further research on these relationships will require larger datasets and more advanced modeling techniques to explore not only these relationships, but also sequential or reciprocal effects. However, despite these constraints, the present findings are useful in delineating how happiness and psychological safety separately mediate the CSR perceptions-burnout relationship, which provides a good basis for developing this line of enquiry in future studies.

## 6. Conclusions

Our research tried to bridge the gap between theoretical concepts and practical implications within the context of healthcare by showing how employees' CSR perceptions can contribute to global SDG framework, as far as China is concerned. The present study has examined how certain concepts such as altruism, happiness, job burnout, and psychological safety are related to CSR perceptions. Thereby, indicating that CSR programs could alleviate job burnout while reliance on motivations and welfare roles by organizations should concentrate on putting them into effectuality. Likewise, the targeted CSR initiatives have the potential to create a sound working environment thereby contributing to SDG 3 and SDG 8. Our research demonstrates that strategic CSR practices in healthcare are effective tools for improved employee outcomes implying that policymakers, organizational leaders, and stakeholders must comprehend that the benefits from engaging in CSR go beyond individual organizations but also relate to global agendas under SDGs. This indicates clearly why it is necessary for the health sector to integrate its strategic objectives with respect to addressing particular issues such as employee burnout critically while playing its part jointly towards creating a more sustainable as well as equitable future based on the integration of CSR.

Consequently, the present research establishes an overall understanding of the significant relationship between CSR perceptions and employee health in the Chinese healthcare industry. It was established that CSR perceptions have lowered employee burnout and improved their happiness and psychological wellbeing contrary to the Positive Psychology Theory.. The moderating effect of altruism reveals that altruistic employees are more likely to benefit from CSR activities. These findings have practical implications: managers should incorporate CSR into their main strategic plans for creating organizational climates that are conducive to CSR, and government should encourage CSR for maintaining business structures that are sustainable. However, this study is limited to the urban setting and cross-sectional design; thus, it is recommended to conduct future research in the rural setting, using a longitudinal approach and considering cultural differences. Our work contributes to the literature on CSR and employee well-being, providing practical knowledge on how to enhance organizational relations and create a healthy business environment for long-term success.

**Informed Consent Statement:** Informed consent was obtained from each respondent.

## Supporting information

**S1 File. Raw data file sustainable care.**
(TXT)

**S2 File. Human subjects research checklist.**
(DOCX)

**S3 File. Similiarity report (Turnitin).**
(PDF)

## Acknowledgments

The authors are also pleased to acknowledge the financial support from Instituto Politécnico de Setúbal.

## Author contributions

**Conceptualization:** Qinghua Fu, Belal Mahmoud AlWadi.

**Formal analysis:** Qinghua Fu, Rui Dias.

**Methodology:** Belal Mahmoud AlWadi, Matac Liviu Marian.

**Project administration:** Rui Dias.

**Software:** Belal Mahmoud AlWadi, Matac Liviu Marian.

**Writing – original draft:** Qinghua Fu.

**Writing – review & editing:** Belal Mahmoud AlWadi, Rui Dias.

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
