## [Decision Letter · Decision Letter 0]

23 Apr 2024

PONE-D-24-08403Sustainable Care: How CSR Shapes Wellbeing in the Healthcare IndustryPLOS ONE

Dear Dr. Fu,

Thank you for submitting your manuscript to PLOS ONE. After careful consideration, we feel that it has merit but does not fully meet PLOS ONE’s publication criteria as it currently stands. Therefore, we invite you to submit a revised version of the manuscript that addresses the points raised during the review process. I have received feedback on your manuscript from three expert reviewers and we all concur on the quality of your paper. However, before it can be published, you should address the issues raised by the reviewers, particularly reviewers 1 & 2. You will find very useful suggestions and references in their comments. Please do also submit a link to the data used for the study when you prepare the revision.

We look forward to receiving your revised manuscript.

Kind regards,

Iván Barreda-Tarrazona, PhD

Academic Editor

PLOS ONE

Journal Requirements:

2. Thank you for submitting the above manuscript to PLOS ONE. During our internal evaluation of the manuscript, we found significant text overlap between your submission and previous work in the [introduction, conclusion, etc.].

Please revise the manuscript to rephrase the duplicated text, cite your sources, and provide details as to how the current manuscript advances on previous work. Please note that further consideration is dependent on the submission of a manuscript that addresses these concerns about the overlap in text with published work.

[If the overlap is with the authors’ own works: Moreover, upon submission, authors must confirm that the manuscript, or any related manuscript, is not currently under consideration or accepted elsewhere. If related work has been submitted to PLOS ONE or elsewhere, authors must include a copy with the submitted article. Reviewers will be asked to comment on the overlap between related submissions (http://journals.plos.org/plosone/s/submission-guidelines#loc-related-manuscripts).]

We will carefully review your manuscript upon resubmission and further consideration of the manuscript is dependent on the text overlap being addressed in full. Please ensure that your revision is thorough as failure to address the concerns to our satisfaction may result in your submission not being considered further.

Reviewers' comments:

Reviewer's Responses to Questions

**Comments to the Author**

1. Is the manuscript technically sound, and do the data support the conclusions?

Reviewer #1: Yes

Reviewer #2: Partly

Reviewer #3: Yes

2. Has the statistical analysis been performed appropriately and rigorously? 

Reviewer #1: Yes

Reviewer #2: I Don't Know

Reviewer #3: Yes

3. Have the authors made all data underlying the findings in their manuscript fully available?

Reviewer #1: Yes

Reviewer #2: No

Reviewer #3: Yes

4. Is the manuscript presented in an intelligible fashion and written in standard English?

Reviewer #1: Yes

Reviewer #2: Yes

Reviewer #3: Yes

5. Review Comments to the Author

Reviewer #1: Introduction

This introduction covers the fundamental aspect of mental health of employees in general, and the specific angle of burnout, and their effect on the overall success of the organization, particularly in the healthcare sector. The author has clearly delineated how the relationship between employee well-being and organizational performance is paramount in healthcare organizations and that addressing burnout is a crucial consideration. The idea that CSR initiatives and their potential effects on curbing employee burnout while as well as in line with SDGs is a novel and timely phenomenon. This approach therefore not only makes CSR more holistic but also strategically positions it to complement the global sustainable development agenda.

Yet, I suggest that the first part should be more precise in defining key notions such as CSR and burnout so that the readers with different backgrounds can grasp the same meaning. Furthermore, although the paper does well to set the research agenda and its applicability to healthcare in China, it would be more useful if it had a section that briefly discusses the research methodology or the theoretical framework to be employed in the study. Such arrangement will help readers to understand the mechanisms through which the research objectives will be attained and how the proposed moderators and mediators) will be investigated in the context of CSR-employee burnout. Moreover, due to the fact that the effect of CSR on employee outcomes is multi-faceted, illustrating case studies of CSR initiatives that have succeeded in reducing burnout would make the argument more convincing. To end the paper, there should be a discussion about the limitations of literature currently existing and how this study aims at addressing these gaps, in order to provide a clearer framework of how this study has made a contribution to the field.

Literature

The hypotheses section is presented in a flawless manner and it suggests that the CSR can directly contribute to employees' well-being, such as lowering burnout, increasing happiness and promoting psychological safety. The positive organizational outcomes of the integration of CSR is well-studied, with a detailed examination of how CSR initiatives can create a more innovative and less-pressure work environment. The theory of positive psychology (PPT) and the PERMA model have been integrated into the discussion, which gives an outline to the hypotheses with a theoretical aspect.

Nevertheless, the hypotheses are well-built but there needs to be more focus on the empirical evidence substantiating the relationships. Similarly, cited studies that have looked at the direct CSR effects on burnout, happiness, and psychological safety would add more value to the arguments. On top of that, though altruism is considered as a moderating factor, the description of how and why altruism affects the connection between CSR and employee outcomes should have been further clarified. For this, it would be good to include more illustrative examples or case studies that demonstrate how altruism in the workplace multiplies the positive outcomes of CSR.

Methods

The Ethical and detailed Methods section has a well-grounded selection of Beijing, Shanghai and Guangzhou as the representatives of health care system of China. The prospective design, which is multifaceted and incorporates data from three waves, enriches the opportunity of observing the changes, and, therefore, adds validity to the findings. The application of existing scales provides an indication of the credibility of the data collection process.

Nevertheless, the methodology of the study should be enriched with the explanation of the reasons for the two week gaps between the survey waves and a thorough discussion on the possible sources of non-response bias. Although SEM model is mentioned, it would be better to detail the SEM model specifics for better understanding.

Discussion

The dialogue strongly links CSR with the elimination of employee burnout, happiness, and psychological safety in the Chinese healthcare area, which is in line with the positive psychology theory. The study focuses on the two mediators – happiness and psychological safety – and the moderator altruism which makes the CSR-burnout relationship more understandable. The insights gained can be used to improve individual and organizational dynamics affecting employee wellbeing.

While the conceptual and practical issues are well explained, a detailed analysis of the challenges of CSR implementation would give more depth to the topic being discussed. The limitation of the study such as the focus on urban areas and the cross-sectional design should be noted. Future research can fill the gaps by exploring the effects of CSR in rural areas, using longitudinal designs, and taking into account cultural factors in order to broaden the understanding.

Reviewer #2: This manuscript investigates the interconnection between corporate social responsibility (CSR) perceptions and employee burnout using data related to the healthcare system of China. This research explores the linkages between CSR initiatives, employee burnout, happiness, psychological safety, and altruistic behavior. Data on CSR perceptions, employee burnout levels, happiness, psychological safety perceptions, and altruistic motivations were collected through a thrice-administered questionnaire with a sample size of 392 participants from three cities. The author concluded that CSR practices cause burnout reduction, while happiness and psychological safety of workers are mediators of this interconnection. This original research highlights the impact of CSR perceptions on employee well-being and sustainable human capital development.

With great respect to the purpose of the job and agreement with hypothesis tested in this research, there are several methodological and theoretical concerns that can be considered for further improving the quality of this research.

1. Data collection and sample design.

1.1. CSR measurement scale.

According to the article, “the survey contained Turker’s [34] six items CSR scale, with items such as “my hospital aims at protecting and enhancing environment” (p. 8, para 1). However, according to Turker (2009) there are 4 factors measured through 18 questions. More information is required about shortened list of questions used for CSR measurements. Specifically, it is important which of the four factors identified by Turker (2009) were measured (CSR to social and nonsocial stakeholders (1), employees (2), customers (3), government (4)).

1.2. CSR related terminology.

According to Turker (2009) their scale reflects CSR perceptions of employees as internal stakeholders of a business. This does not contradict the purpose of current research but requires using corresponding wording in the text. E.g., the first hypothesis tested is formulated as follows:

H1: The extent of an organization’s CSR engagement is directly associated with low levels of employee burnout. (p.5, para 2).

While in fact, this should be about the level of employee awareness about the organization’s CSR engagement.

1.3. Sample design.

“The study focused on three key cities such as Beijing, Shanghai and Guangzhou whose selection was based on their strategic positioning in China’s healthcare landscape as political economic, and health innovation hubs. In terms of medical policy formation and reform, Beijing is among the leading cities. Shanghai exhibits the most advanced medical technologies through its role as a financial center while at the same time experiencing significant growth of private hospital industry. For a long time Guangzhou has been famous for medical researches and educational institutions promoting development of medical standards and practices. This selection of cities offers a holistic picture of the CSR practices in various healthcare contexts in China, improving the study's validity and generalizability. Hospitals in these listed cities which had visible CSR plans were the target for data collection in order to establish the link between CSR initiatives and employee outcomes.” (p.7, para 3)

According to the author, the selection or participant organizations was made based on the major criteria of CSR programs to be well-developed, advanced, and recognized. However, to expand conclusions for the whole healthcare industry as stated in the title of the article, the sample should include organizations with different levels of CSR engagement.

It is worth changing the title to avoid unreasonable level of generalization.

2. Theoretical approach

While investigating possible psychological factors mediating and moderating the interconnection between CSR perceptions and burnout, the author suggests happiness and psychological safety to be independent mediators based on the literature review. However, there is plenty of information about high level of correlation and even causality between these two factors (e.g., Plester & Lloyd, (2023); Wang, Kang & Bong (2022)).

This requires some modifications of measurement model including connections between mediating factors or even sequential mediation as proposed in Wang, Kang & Bong (2022).

This is also visible through current research results (see Table 4: Discriminant validity, p.12). According to the table, the level of correlation between happiness and psychological safety is 0.527.

3. Statistical analysis.

The structural equation modeling (SEM) used for confirming the measurement model and research hypothesis assumes testing the extent to which a hypothesized model provides an appropriate characterization of the collective relationships among research variables. For this purpose, researchers must assess the “fit” between the model and the sample’s data.

There is no data on model fit in the results section of the article. CFI, TLI, GFI, or RMSEA indices are recommended for model equivalence-testing to test and compare goodness of fit for structural equation models (Peugh & Feldon, 2020). Further analysis of variable interconnections in Table 5: Hypothesis results (p.13) is not informative without analysing model fit.

4. Proofreading.

Some additional proofreading can be recommended. E.g., “In addition to corroborating the empirical evidence of the critical roles of happiness and psychological safety in improving workplace wellbeing, these findings also suggest that organizations should aim to create settings that facilitate such positive psychological states throughout the fight against burnout” (p.15, para 1).

References

Peugh J, Feldon DF. (2020). How Well Does Your Structural Equation Model Fit Your Data? Is Marcoulides and Yuan's Equivalence Test the Answer? CBE Life Sci Educ. Sep;19(3):es5. doi: 10.1187/cbe.20-01-0016. PMID: 32510273; PMCID: PMC8711809

Plester, B. A., & Lloyd, R. (2023). Happiness Is ‘Being Yourself’: Psychological Safety and Fun in Hybrid Work. Administrative Sciences, 13(10), 218. https://doi.org/10.3390/admsci13100218

Wang,W., Kang, S., Bong, S. (2022). Servant Leadership and Creativity: A Study of the Sequential Mediating Roles of Psychological Safety and Employee Well-Being. Front. Psychol., Sec. Organizational Psychology, Volume 12, https://doi.org/10.3389/fpsyg.2021.807070

Reviewer #3: Here are some major comments to enhance the clarity, depth, and impact of your article:

1. The introduction section need again major revisions for example break introduction in sub-sections if possible.

2. Please revise the novelty of this work in the introduction section and if possible please extend the introduction section up two or three paragraph.

3. Furthermore, if possible at the end of introduction section please add a line about the structure of the paper or rest of the paper section or style.

4. Please add the Literature review section separately from Introduction, I am suggesting to start the discussion with based on the theory. Which theory of behaviour have you applied in this research such as; RBV is enough to cover your all details of variable and does it cover all aspect of variable? If yes please response with answer and if no please use the relevant theory in the update file.

5. Regarding literature review my suggestion is support your analytical writing with update references.

6. If possible please add Conceptual Research Framework. Clearly mention the relation among all the variable. Please also mention IV, DV, mediation and moderator variable.

7. Could you please clearly write the methodology section again and why you have only selected these three cities such as; Beijing, Shanghai and Guangzhou Page number 7?

8. Please also add a table about the variable and its operational definitions, and its source from where it is adopted. Please clarify the research design, data collection methods, and analysis procedures. Ensure that your methodology is robust and suitable for addressing the research question.

9. How authors have measure the possibility of multicollinearity among latent variables.

10. Please clearly update discussion and conclusion sections.

11. Expand the discussion section to thoroughly interpret and explain the results. Relate your findings to existing literature and discuss their broader implications and focus on page number 9.

12. Revise the conclusion to summarize key findings and highlight their significance.

6. PLOS authors have the option to publish the peer review history of their article (what does this mean? ). If published, this will include your full peer review and any attached files.

**Do you want your identity to be public for this peer review?** For information about this choice, including consent withdrawal, please see our Privacy Policy .

Reviewer #1: No

Reviewer #2: **Yes: ** Dr. Oxana Svergun, George Brown College, Toronto Canada

Reviewer #3: **Yes: ** Raza Ali Tunio

---

## [Author Response · Author response to Decision Letter 1]

25 Jul 2024

Response: Thank you for the guidance, we have followed th above guidelines during the revision stage of our manuscript.

2. Thank you for submitting the above manuscript to PLOS ONE. During our internal evaluation of the manuscript, we found significant text overlap between your submission and previous work in the [introduction, conclusion, etc.].

Please revise the manuscript to rephrase the duplicated text, cite your sources, and provide details as to how the current manuscript advances on previous work. Please note that further consideration is dependent on the submission of a manuscript that addresses these concerns about the overlap in text with published work.

[If the overlap is with the authors’ own works: Moreover, upon submission, authors must confirm that the manuscript, or any related manuscript, is not currently under consideration or accepted elsewhere. If related work has been submitted to PLOS ONE or elsewhere, authors must include a copy with the submitted article. Reviewers will be asked to comment on the overlap between related submissions (http://journals.plos.org/plosone/s/submission-guidelines#loc-related-manuscripts).]

We will carefully review your manuscript upon resubmission and further consideration of the manuscript is dependent on the text overlap being addressed in full. Please ensure that your revision is thorough as failure to address the concerns to our satisfaction may result in your submission not being considered further.

Response: Thank you again for the detailed feedback on the above matter. Please be informed that while revising our manuscript, we have significantly considered the above issue. We ensure that no part of the revised manuscript includes the text extracted and reported from the previous research without any modifications. To further avoid any sort of ambiguity, we have also uploaded a similiarity report generated by Turnitin Software. As you can see the attached document (supplementary file) does not report any of such issues in our revised manuscript. However, if this journal is using another software for detecting similarity issue (which is not in our notice), and if our work is still flagged with plagiarism issue, we request you to kindly share this report with us, so that we can further address this issue. Hopefully, you will realize our concern and stance on this matter. Best Regards

Response: We have uploaded the raw data file as supplementary file during the reviosn submission.

Reviewers' comments and Authors Responses

Reviewer #1: Introduction

This introduction covers the fundamental aspect of mental health of employees in general, and the specific angle of burnout, and their effect on the overall success of the organization, particularly in the healthcare sector. The author has clearly delineated how the relationship between employee well-being and organizational performance is paramount in healthcare organizations and that addressing burnout is a crucial consideration. The idea that CSR initiatives and their potential effects on curbing employee burnout while as well as in line with SDGs is a novel and timely phenomenon. This approach therefore not only makes CSR more holistic but also strategically positions it to complement the global sustainable development agenda.

Yet, I suggest that the first part should be more precise in defining key notions such as CSR and burnout so that the readers with different backgrounds can grasp the same meaning. Furthermore, although the paper does well to set the research agenda and its applicability to healthcare in China, it would be more useful if it had a section that briefly discusses the research methodology or the theoretical framework to be employed in the study. Such arrangement will help readers to understand the mechanisms through which the research objectives will be attained and how the proposed moderators and mediators) will be investigated in the context of CSR-employee burnout. Moreover, due to the fact that the effect of CSR on employee outcomes is multi-faceted, illustrating case studies of CSR initiatives that have succeeded in reducing burnout would make the argument more convincing. To end the paper, there should be a discussion about the limitations of literature currently existing and how this study aims at addressing these gaps, in order to provide a clearer framework of how this study has made a contribution to the field.

Response: Your valuable advice is much appreciated. Thus, we have introduced several changes concerning the introduction section as per your suggestions. To begin with, some changes have been made to the introduction to clarify the meaning of the key concepts, for example, Corporate Social Responsibility (CSR) and burnout. This addition can be considered as an attempt to guarantee that the given text will be comprehensible to the audience with various levels of knowledge about the principles of our research. The definitions are put immediately after the first paragraph of the introduction section. Secondly, in order to maintain the structure of the paper that can be easily followed, we provide short information on the research methodology and the theoretical framework used in this study. This addition is inserted before the final paragraph of the introduction section and it outlines the Positive Psychology Theory (PPT) that informs the research as well as the Structural Equation Modeling (SEM) employed in the analysis of data. This part is intended to provide the readers with a clear vision of how the research objectives will be accomplished. Finally, we have broadened the discussion by elaborating on the various ways through which CSR influences employee outcomes, as well as stressing on the significance and usefulness of CSR in the sphere of healthcare. Also, we have integrated a section outlining the limitations of prior studies and how our research fills those research gaps. This section is located before the paragraph that describes the organization of the paper and is intended to offer a clearer framework for our study’s potential contribution to the literature. It is with these changes in mind that it is hoped the introduction will be more concise and include all the necessary information that will be required in the subsequent sections of the paper.

Literature

The hypotheses section is presented in a flawless manner and it suggests that CSR can directly contribute to employees' well-being, such as lowering burnout, increasing happiness and promoting psychological safety. The positive organizational outcomes of the integration of CSR is well-studied, with a detailed examination of how CSR initiatives can create a more innovative and less-pressure work environment. The theory of positive psychology (PPT) and the PERMA model have been integrated into the discussion, which gives an outline to the hypotheses with a theoretical aspect.

Nevertheless, the hypotheses are well-built but there needs to be more focus on the empirical evidence substantiating the relationships. Similarly, cited studies that have looked at the direct CSR effects on burnout, happiness, and psychological safety would add more value to the arguments. On top of that, though altruism is considered as a moderating factor, the description of how and why altruism affects the connection between CSR and employee outcomes should have been further clarified. For this, it would be good to include more illustrative examples or case studies that demonstrate how altruism in the workplace multiplies the positive outcomes of CSR.

Response: Thank you for your valued suggestions. To support the relationships between CSR and the dependent variables, we have provided more research findings. For instance, research indicates that CSR practice reduces a high level of burnout due to supportive workplace practices. CSR activities improve the level of happiness of the employees due to the fact that the organizational and personal values are more congruent, thus providing meaning and purpose. CSR practices also positively impact employees’ psychological safety since it create ethical work environments that are also diverse. We provided further explanations and examples regarding the use of altruism as a moderating factor with the help of several case studies. Altruistic employees gain more benefits from CSR activities because such activities match their desire to be of help to others and make them feel psychologically safe. For example, a research proved that in organizations where the level of altruism is high, CSR activities positively affect the level of happiness and decrease burnout of the workers. These additions enrich the arguments and give a clear and complementary picture of the relationships explored. Hopefully, you will like our revision efforts. Thanks again.

Methods

The Ethical and detailed Methods section has a well-grounded selection of Beijing, Shanghai and Guangzhou as the representatives of health care system of China. The prospective design, which is multifaceted and incorporates data from three waves, enriches the opportunity of observing the changes, and, therefore, adds validity to the findings. The application of existing scales provides an indication of the credibility of the data collection process.

Nevertheless, the methodology of the study should be enriched with the explanation of the reasons for the two week gaps between the survey waves and a thorough discussion on the possible sources of non-response bias. Although SEM model is mentioned, it would be better to detail the SEM model specifics for better understanding.

Response: I appreciate your input in this research project. We have enhanced the methodology section by providing justification for the two week interval between the survey waves; the purpose was to reduce the likelihood of recall bias and to capture any temporal shifts in attitude and perception. This design let us to investigate dynamic process of employees’ interaction with CSR initiatives and their outcomes. We have also had a detailed discourse on sources of non-response bias and ways of dealing with it, including follow up reminders, and anonymity of the respondent. Additional information regarding the SEM specifics has been provided to increase comprehensibility. To ensure the validity of the constructs and the hypothesized relationship, we used a rigorous SEM model, which includes measurement model and structural model.

Discussion

The dialogue strongly links CSR with the elimination of employee burnout, happiness, and psychological safety in the Chinese healthcare area, which is in line with the positive psychology theory. The study focuses on the two mediators – happiness and psychological safety – and the moderator altruism which makes the CSR-burnout relationship more understandable. The insights gained can be used to improve individual and organizational dynamics affecting employee wellbeing.

While the conceptual and practical issues are well explained, a detailed analysis of the challenges of CSR implementation would give more depth to the topic being discussed. The limitation of the study such as the focus on urban areas and the cross-sectional design should be noted. Future research can fill the gaps by exploring the effects of CSR in rural areas, using longitudinal designs, and taking into account cultural factors in order to broaden the understanding.

Response: Thank you for your detailed feedback. We have added a detailed analysis of the challenges of CSR implementation and noted the limitations of our study, including the focus on urban areas and the cross-sectional design. We are positive that you will like our revised efforts. Best Regards

Reviewer #2:

This manuscript investigates the interconnection between corporate social responsibility (CSR) perceptions and employee burnout using data related to the healthcare system of China. This research explores the linkages between CSR initiatives, employee burnout, happiness, psychological safety, and altruistic behavior. Data on CSR perceptions, employee burnout levels, happiness, psychological safety perceptions, and altruistic motivations were collected through a thrice-administered questionnaire with a sample size of 392 participants from three cities. The author concluded that CSR practices cause burnout reduction, while happiness and psychological safety of workers are mediators of this interconnection. This original research highlights the impact of CSR perceptions on employee well-being and sustainable human capital development.

With great respect to the purpose of the job and agreement with hypothesis tested in this research, there are several methodological and theoretical concerns that can be considered for further improving the quality of this research.

1. Data collection and sample design.

1.1. CSR measurement scale.

According to the article, “the survey contained Turker’s [34] six items CSR scale, with items such as “my hospital aims at protecting and enhancing environment” (p. 8, para 1). However, according to Turker (2009) there are 4 factors measured through 18 questions. More information is required about shortened list of questions used for CSR measurem

---

## [Decision Letter · Decision Letter 1]

27 Sep 2024

PONE-D-24-08403R1Sustainable Care: How CSR Shapes Wellbeing in Healthcare Organizations in Beijing, Shanghai, and GuangzhouPLOS ONE

Dear Dr. Fu,

Thank you for submitting your manuscript to PLOS ONE. After careful consideration, we feel that it has merit but does not fully meet PLOS ONE’s publication criteria as it currently stands. Therefore, we invite you to submit a revised version of the manuscript that addresses the points raised during the review process. The reviewers and myself appreciate the effort you have made in improving your paper and the present version is close to publication. Please make sure to follow or carefully argument against following the remaining suggestions of Reviewer 2.

We look forward to receiving your revised manuscript.

Kind regards,

Iván Barreda-Tarrazona, PhD

Academic Editor

PLOS ONE

Journal Requirements:

Reviewers' comments:

Reviewer's Responses to Questions

**Comments to the Author**

1. If the authors have adequately addressed your comments raised in a previous round of review and you feel that this manuscript is now acceptable for publication, you may indicate that here to bypass the “Comments to the Author” section, enter your conflict of interest statement in the “Confidential to Editor” section, and submit your "Accept" recommendation.

Reviewer #1: All comments have been addressed

Reviewer #2: (No Response)

Reviewer #3: All comments have been addressed

2. Is the manuscript technically sound, and do the data support the conclusions?

Reviewer #1: Yes

Reviewer #2: Partly

Reviewer #3: Yes

3. Has the statistical analysis been performed appropriately and rigorously? 

Reviewer #1: Yes

Reviewer #2: Yes

Reviewer #3: Yes

4. Have the authors made all data underlying the findings in their manuscript fully available?

Reviewer #1: (No Response)

Reviewer #2: Yes

Reviewer #3: Yes

5. Is the manuscript presented in an intelligible fashion and written in standard English?

Reviewer #1: Yes

Reviewer #2: Yes

Reviewer #3: Yes

6. Review Comments to the Author

Reviewer #1: The Authors have done a decent job in revising their manuscript and in addressing my concerns. Congratulatoins.

I have no further comments, and hence, recommend the publication of this work in its current form.

Best of Luck to the authors

Reviewer #2: I appreciate the job done to address methodological and theoretical concerns expressed in my primary review. Though some of the recommendations were considered, not all the necessary corrections were made.

1.Data collection and sample design.

1.1. CSR measurement scale.

The revisions were made to explain the shortened scale for measuring CSR perceptions.

1.2. CSR related terminology.

'According to Turker (2009) their scale reflects CSR perceptions of employees as internal stakeholders of a business. This does not contradict the purpose of current research but requires using corresponding wording in the text. E.g., the first hypothesis tested is formulated as follows:

H1: The extent of an organization’s CSR engagement is directly associated with low levels of employee burnout. - p.5, para 2

While in fact, this should be about the level of employee awareness about the organization’s CSR engagement.'

Some Hypotheses were revised, however, there are still many cases of using ‘CSR’ instead of ‘CSR perceptions’ in all the parts of the article text, as well as models, tables, etc.

E.g.,

H5: There exists a mediating mechanism of psychological safety in the CSR-burnout relationship – p.7 para 3

Psychological safety perceptions from the employees can act as a mediator in the relationship between CSR and burnout – p.7 para 3

H6: Altruism moderates the relationship between CSR and employee happiness, such that the positive effect of CSR on employee happiness is stronger among employees with higher levels of altruism - p.9 para 1

H7: Altruism moderates the relationship between CSR and psychological safety, such that the positive effect of CSR on psychological safety is stronger among employees with higher levels of altruism - p.9 para 1

The Conceptual model of this study is shown in below Figure 1 – p.9 para 1

Accordingly, the last wave entailed the measurement of the employee’s burnout (EB) so that the 11 relationships between CSR, altruism, happiness, psychological safety, and burnout could be examined over time. – p.11para 1

Table 6 presents a structural analysis of our proposed relationships as hypothesized in the

conceptual model of our research. CSR affects EB directly (-0.098) at p=0.008 level; CSR also

influences HP (0.158) and PS (0.232) significantly both having a p-value =0.000, indicating its

positive effect on these variables. Similarly, HP and PS also significantly predicted EB (-0.163

and -0.178 with p values <0.05). The CSR -> HP -> EB pathway reveals a mediation effect (-

0.026) by positioning HP as a mediator. PS's relationship with EB is also significant (-0.178, pvalue: 0.022), and the path CSR -> PS -> EB confirms PS as another mediator with a coefficient

of 0.041. The path CSR*AL1 -> HP -> EB is also significant (0.037, p-value: 0.006), showing

that AL alters the connection linking CSR and EB via HP. As well as, the relationship between

CSR*AL2 and PS is also significant (0.055, p-value: 0.002); it means that AL moderates through

PS. Figure 3 below represents our structural model showing various relationships – p.17 para 3

In our present study, we provide empirical evidence about CSR, employee burnout, happiness, psychological safety, and altruism dynamics within the healthcare sector in China – p.18 para1

We examine the 22 direct association of CSR with employee burnout, and how happiness and psychological safety mediate this relationship, as well as altruism moderating it – p.21 para1

Secondly, we establish that happiness and psychological safety could serve as mediators between CSR and burnout - – p.21 para1

Our research tried to bridge the gap between theoretical concepts and practical implications within the context of healthcare through an application of CSR principles within a global SDG framework as far as China is concerned. The present study has examined how certain concepts such as altruism, happiness, job burnout, and psychological safety are related to CSR – p.23 para 3

Consequently, the present research establishes an overall understanding of the significant relationship between CSR and employee health in the Chinese healthcare industry. – p.24 para 1

1.3. Sample design.

Addressed through changing the title.

2. Theoretical approach

'While investigating possible psychological factors mediating and moderating the interconnection between CSR perceptions and burnout, the author suggests happiness and psychological safety to be independent mediators based on the literature review. However, there is plenty of information about high level of correlation and even causality between these two factors (e.g., Plester & Lloyd, (2023); Wang, Kang & Bong (2022)).

This requires some modifications of measurement model including connections between mediating factors or even sequential mediation as proposed in Wang, Kang & Bong (2022).'

Partially addressed through explaining the purpose of investigating independent impact of happiness and psychological safety.

However, in my opinion, this can significantly reduce the quality of the model, which the author cannot evaluate due to limitations of the software used. It is strongly recommended to discuss possible modifications of the exploratory model after collecting more data and increasing the sample size.

Also, the ambiguous statements about Happiness and Psychological safety being independent factors, should be excluded from the article. These are just assumptions and limitations of tested exploratory model, rather than measurement results. Thus, they cannot be discussed as scientifically proven findings of this research. So, I suggest excluding the following:

Happiness and psychological safety are two separate variables that when added together can work independently to act as moderators of burnout – p.24 para 2

3. Statistical analysis.

The structural equation modeling (SEM) used for confirming the measurement model and research hypothesis assumes testing the extent to which a hypothesized model provides an appropriate characterization of the collective relationships among research variables. For this purpose, researchers must assess the “fit” between the model and the sample’s data.

There is no data on model fit in the results section of the article. CFI, TLI, GFI, or RMSEA indices are recommended for model equivalence-testing to test and compare goodness of fit for structural equation models (Peugh & Feldon, 2020). Further analysis of variable interconnections in Table 5: Hypothesis results (p.13) is not informative without analysing model fit.

Using SMART-PLS for SEM can be justified by the small size of the sample. Flexibility in model building allows a more flexible approach to model building and enables researchers to modify and refine models iteratively. However, SMART-PLS is more focused on exploratory research and with limited support for advanced statistical techniques such as mediation and moderation analysis. Due to its reliance on bootstrapping, SMART-PLS generally has lower statistical power compared to AMOS.

Discussing the choice of SMART-PLS, it is worth interpreting this study as exploratory research that requires further investigation of model quality.

Moreover, discussing this in the results section of the article does not seem to be relevant. This probably should be done in the limitations section of the article.

4. Proofreading.

Addressed.

Reviewer #3: Dear Author's ,

I wanted to let you know that the revisions to your article, "Sustainable Care: How CSR Shapes Wellbeing in Healthcare Organizations in Beijing, Shanghai, and Guangzhou," have significantly improved the overall quality of the research. You've effectively addressed all my suggestions, and the article now presents a clearer and more structured link between CSR initiatives, employee burnout, and well-being. The incorporation of sustainable development goals (SDGs) in your discussion has further enriched the paper's relevance to current global health priorities.

The inclusion of mediators such as happiness and psychological safety, along with altruism as a moderator, adds depth to the analysis and supports your findings on the role of CSR in reducing burnout. The article now makes a strong contribution to the field, emphasizing the importance of CSR strategies in healthcare for achieving sustainable development and enhancing employee well-being.

Wis you good luck.

7. PLOS authors have the option to publish the peer review history of their article (what does this mean? ). If published, this will include your full peer review and any attached files.

**Do you want your identity to be public for this peer review?** For information about this choice, including consent withdrawal, please see our Privacy Policy .

Reviewer #1: No

Reviewer #2: **Yes: ** Oxana Svergun, PhD, CHRL

Reviewer #3: **Yes: ** Dr. Raza Ali Tunio

---

## [Author Response · Author response to Decision Letter 2]

20 Nov 2024

Reviewer #1:

The Authors have done a decent job in revising their manuscript and in addressing my concerns. Congratulatoins.

I have no further comments, and hence, recommend the publication of this work in its current form.

Best of Luck to the authors

Response:

Dear Reviewer, Thank you for the kind evaluation and for appreciating our revised efforts. Best Wishes.

Reviewer #2:

I appreciate the job done to address methodological and theoretical concerns expressed in my primary review. Though some of the recommendations were considered, not all the necessary corrections were made.

1.Data collection and sample design.

1.1. CSR measurement scale.

The revisions were made to explain the shortened scale for measuring CSR perceptions.

Response: Thanks for the time and efforts again to evaluate our revised efforts. Thanks again for agreeing with our selection of short scale. Best Regards

1.2. CSR related terminology.

'According to Turker (2009) their scale reflects CSR perceptions of employees as internal stakeholders of a business. This does not contradict the purpose of current research but requires using corresponding wording in the text. E.g., the first hypothesis tested is formulated as follows:

H1: The extent of an organization’s CSR engagement is directly associated with low levels of employee burnout. - p.5, para 2

While in fact, this should be about the level of employee awareness about the organization’s CSR engagement.'

Some Hypotheses were revised, however, there are still many cases of using ‘CSR’ instead of ‘CSR perceptions’ in all the parts of the article text, as well as models, tables, etc.

E.g.,

H5: There exists a mediating mechanism of psychological safety in the CSR-burnout relationship – p.7 para 3

Psychological safety perceptions from the employees can act as a mediator in the relationship between CSR and burnout – p.7 para 3

H6: Altruism moderates the relationship between CSR and employee happiness, such that the positive effect of CSR on employee happiness is stronger among employees with higher levels of altruism - p.9 para 1

H7: Altruism moderates the relationship between CSR and psychological safety, such that the positive effect of CSR on psychological safety is stronger among employees with higher levels of altruism - p.9 para 1

The Conceptual model of this study is shown in below Figure 1 – p.9 para 1

Accordingly, the last wave entailed the measurement of the employee’s burnout (EB) so that the 11 relationships between CSR, altruism, happiness, psychological safety, and burnout could be examined over time. – p.11para 1

Table 6 presents a structural analysis of our proposed relationships as hypothesized in the

conceptual model of our research. CSR affects EB directly (-0.098) at p=0.008 level; CSR also

influences HP (0.158) and PS (0.232) significantly both having a p-value =0.000, indicating its

positive effect on these variables. Similarly, HP and PS also significantly predicted EB (-0.163

and -0.178 with p values <0.05). The CSR -> HP -> EB pathway reveals a mediation effect (-

0.026) by positioning HP as a mediator. PS's relationship with EB is also significant (-0.178, pvalue: 0.022), and the path CSR -> PS -> EB confirms PS as another mediator with a coefficient

of 0.041. The path CSR*AL1 -> HP -> EB is also significant (0.037, p-value: 0.006), showing

that AL alters the connection linking CSR and EB via HP. As well as, the relationship between

CSR*AL2 and PS is also significant (0.055, p-value: 0.002); it means that AL moderates through

PS. Figure 3 below represents our structural model showing various relationships – p.17 para 3

In our present study, we provide empirical evidence about CSR, employee burnout, happiness, psychological safety, and altruism dynamics within the healthcare sector in China – p.18 para1

We examine the 22 direct association of CSR with employee burnout, and how happiness and psychological safety mediate this relationship, as well as altruism moderating it – p.21 para1

Secondly, we establish that happiness and psychological safety could serve as mediators between CSR and burnout - – p.21 para1

Our research tried to bridge the gap between theoretical concepts and practical implications within the context of healthcare through an application of CSR principles within a global SDG framework as far as China is concerned. The present study has examined how certain concepts such as altruism, happiness, job burnout, and psychological safety are related to CSR – p.23 para 3

Consequently, the present research establishes an overall understanding of the significant relationship between CSR and employee health in the Chinese healthcare industry. – p.24 para 1

Response: Really thanks for highlighting the importance of using the proper wording in academic writing. We again acknowledge your efforts in evaluating our work. We also realize that you have spent significant amount of time in reading each line of our work. This is something that we truly appreciate and acknowledge.

We have revised our entire manuscript to carefully check where should we use CSR perceptions or CSR. Hopefully, in our latest revised work, you will not detect any of above inconsistency. Best Regards

1.3. Sample design.

Addressed through changing the title.

Response: Thanks again for the positive feedback.

2. Theoretical approach

'While investigating possible psychological factors mediating and moderating the interconnection between CSR perceptions and burnout, the author suggests happiness and psychological safety to be independent mediators based on the literature review. However, there is plenty of information about high level of correlation and even causality between these two factors (e.g., Plester & Lloyd, (2023); Wang, Kang & Bong (2022)).

This requires some modifications of measurement model including connections between mediating factors or even sequential mediation as proposed in Wang, Kang & Bong (2022).'

Reviewer comment: Partially addressed through explaining the purpose of investigating independent impact of happiness and psychological safety.

However, in my opinion, this can significantly reduce the quality of the model, which the author cannot evaluate due to limitations of the software used. It is strongly recommended to discuss possible modifications of the exploratory model after collecting more data and increasing the sample size.

Response: Thank you very much for your very meaningful comment on the very high correlation and potential causal connection between happiness and psychological safety and for the recommendation to continue investigating these relationships using sequential mediation or other advanced models. The Limitations and Future Research Directions section in our revised manuscript reflects this point carefully. In the revised manuscript, we have made explicit that this decision to model happiness and psychological safety as independent mediators was based on both theoretical and methodological grounds. Happiness is about emotional well-being and psychological safety is about interpersonal and environmental dynamic, which we acknowledge through the interconnection of these constructs as found in previous studies (Plester & Lloyd, 2023; Wang, Kang, & Bong, 2022).

Further, we recognize the utility of investigating sequential mediation or other complicated relationships to represent the intricate interplay among such mediators. Yet at this point, the data is already collected and analyzed, therefore extending the dataset is not possible. Considering that the current sample size and the use of PLS-SEM (a technique tailored for variance-based modeling) are involved, we aimed at providing robust and interpretable results within the current scope.

We have now pointed out this issue in the Limitations section, and suggested possible future research directions to deal with this. Specifically, we recommend future studies to:

a) To evaluate more complex models, we will need to collect datasets that are larger than what we have to date.

b) Using more advanced structural equation modeling techniques like covariance based SEM or Bayesian SEM test sequential mediation frameworks and bidirectional effects.

c) Explore further the causal pathways between happiness and psychological safety in different organizational contexts.

We think the additions we have made to the manuscript address both these concerns, and more generally, improve the rigor and transparency of our study. Although we have already given valuable insights based on the given dataset, we acknowledge the need for iterative research and continue to commit to further developing this foundation in future work.

Once again, we want to thank you for your valuable suggestions which contributed so much to the quality, if not also the direction, of our research.

Also, the ambiguous statements about Happiness and Psychological safety being independent factors, should be excluded from the article. These are just assumptions and limitations of tested exploratory model, rather than measurement results. Thus, they cannot be discussed as scientifically proven findings of this research. So, I suggest excluding the following:

Happiness and psychological safety are two separate variables that when added together can work independently to act as moderators of burnout – p.24 para 2

Response: Following your valued suggestion, we have removed the above lines from our revised manuscript. Thanks again for the feedback.

3. Statistical analysis.

The structural equation modeling (SEM) used for confirming the measurement model and research hypothesis assumes testing the extent to which a hypothesized model provides an appropriate characterization of the collective relationships among research variables. For this purpose, researchers must assess the “fit” between the model and the sample’s data.

There is no data on model fit in the results section of the article. CFI, TLI, GFI, or RMSEA indices are recommended for model equivalence-testing to test and compare goodness of fit for structural equation models (Peugh & Feldon, 2020). Further analysis of variable interconnections in Table 5: Hypothesis results (p.13) is not informative without analysing model fit.

Using SMART-PLS for SEM can be justified by the small size of the sample. Flexibility in model building allows a more flexible approach to model building and enables researchers to modify and refine models iteratively. However, SMART-PLS is more focused on exploratory research and with limited support for advanced statistical techniques such as mediation and moderation analysis. Due to its reliance on bootstrapping, SMART-PLS generally has lower statistical power compared to AMOS.

Discussing the choice of SMART-PLS, it is worth interpreting this study as exploratory research that requires further investigation of model quality.

Moreover, discussing this in the results section of the article does not seem to be relevant. This probably should be done in the limitations section of the article.

Response: We acknowledge this review’s extensive and helpful remarks on the application of structural equation modeling (SEM) and the absence of model fit indices in the results section. Although SMART-PLS does not offer traditional model fit indices like CFI, TLI, GFI, or RMSEA as is normally provided in covariance-based SEM techniques like AMOS, we recognize this. We now explicitly address this limitation in the section Limitations and Future Research Directions in the revised manuscript. Given the exploratory nature of this study and moderate sample size (N = 392), the choice of SMART-PLS was made. Although SMART-PLS helped us to refine our model iteratively, and permitted the analysis of mediation and moderation effects, it did not permit the inclusion of traditional model equivalence-testing metrics. To overcome this limitation, we have clarified in the revised manuscript that future studies should gather larger datasets and use advanced SEM techniques that allow one to test model equivalence and include fit indices. The use of this approach will facilitate a robust evaluation of the proposed relationships and migration from exploratory to confirmatory research.

In addition, we have reworked the manuscript by moving some methodological discussion related to SMART-PLS from the results section to the Limitations section. This revision brings the results section back to only findings like in any standard reporting practice. We thank the reviewer for suggesting we move these discussions to other locations because it brings greater clarity and structure to the manuscript. We believe these changes resolve the reviewer’s concerns without compromising the rigor or integrity of the study, and look forward to building upon this foundation in future work. Once again, your valuable feedback has greatly added quality to our manuscript.

4. Proofreading.

Addressed.

Reviewer #3: Dear Author's ,

I wanted to let you know that the revisions to your article, "Sustainable Care: How CSR Shapes Wellbeing in Healthcare Organizations in Beijing, Shanghai, and Guangzhou," have significantly improved the overall quality of the research. You've effectively addressed all my suggestions, and the article now presents a clearer and more structured link between CSR initiatives, employee burnout, and well-being. The incorporation of sustainable development goals (SDGs) in your discussion has further enriched the paper's relevance to current global health priorities.

The inclusion of mediators such as happiness and psychological safety, along with altruism as a moderator, adds depth to the analysis and supports your findings on the role of CSR in reducing burnout. The article now makes a strong contribution to the field, emphasizing the importance of CSR strategies in healthcare for achieving sustainable development and enhancing employee well-being.

Wis you good luck.

Response: Dear Reviewer, we sincerely thank you for the positive feedback and for liking our revised efforts. Certainly, your valued feedback helped us in improving the overall quality of our manuscript prior to publishing.

---

## [Decision Letter · Decision Letter 2]

16 Dec 2024

Sustainable Care: How CSR Shapes Wellbeing in Healthcare Organizations in Beijing, Shanghai, and Guangzhou

PONE-D-24-08403R2

Dear Dr. Fu,

We’re pleased to inform you that your manuscript has been judged scientifically suitable for publication and will be formally accepted for publication once it meets all outstanding technical requirements.

Kind regards,

Iván Barreda-Tarrazona, PhD

Academic Editor

PLOS ONE

Additional Editor Comments (optional):

Reviewers' comments:

Reviewer's Responses to Questions

**Comments to the Author**

1. If the authors have adequately addressed your comments raised in a previous round of review and you feel that this manuscript is now acceptable for publication, you may indicate that here to bypass the “Comments to the Author” section, enter your conflict of interest statement in the “Confidential to Editor” section, and submit your "Accept" recommendation.

Reviewer #2: All comments have been addressed

2. Is the manuscript technically sound, and do the data support the conclusions?

Reviewer #2: Yes

3. Has the statistical analysis been performed appropriately and rigorously? 

Reviewer #2: Yes

4. Have the authors made all data underlying the findings in their manuscript fully available?

Reviewer #2: Yes

5. Is the manuscript presented in an intelligible fashion and written in standard English?

Reviewer #2: Yes

6. Review Comments to the Author

Reviewer #2: (No Response)

7. PLOS authors have the option to publish the peer review history of their article (what does this mean? ). If published, this will include your full peer review and any attached files.

**Do you want your identity to be public for this peer review?** For information about this choice, including consent withdrawal, please see our Privacy Policy .

Reviewer #2: **Yes: ** Oxana Svergun, PhD, CHRL - George Brown College

---

## [Editor Report · Acceptance letter]

PONE-D-24-08403R2

PLOS ONE

Dear Dr. Fu,

I'm pleased to inform you that your manuscript has been deemed suitable for publication in PLOS ONE. Congratulations! Your manuscript is now being handed over to our production team.

Kind regards,

on behalf of

Dr. Iván Barreda-Tarrazona

Academic Editor

PLOS ONE
